# AN ANALYTIC THEORY OF GENERALIZATION DYNAMICS AND TRANSFER LEARNING IN DEEP LINEAR NETWORKS

**Andrew K. Lampinen**
Department of Psychology
Stanford University
lampinen@stanford.edu

**Surya Ganguli**
Department of Applied Physics
Stanford University
and
Google Brain
sganguli@stanford.edu

## ABSTRACT

Much attention has been devoted recently to the generalization puzzle in deep learning: large, deep networks can generalize well, but existing theories bounding generalization error are exceedingly loose, and thus cannot explain this striking performance. Furthermore, a major hope is that knowledge may transfer across tasks, so that multi-task learning can improve generalization on individual tasks. However we lack analytic theories that can quantitatively predict how the degree of knowledge transfer depends on the relationship between the tasks. We develop an analytic theory of the nonlinear dynamics of generalization in deep linear networks, both within and across tasks. In particular, our theory provides analytic solutions to the training and testing error of deep networks as a function of training time, number of examples, network size and initialization, and the task structure and SNR. Our theory reveals that deep networks progressively learn the most important task structure first, so that generalization error at the early stopping time primarily depends on task structure and is independent of network size. This suggests any tight bound on generalization error must take into account task structure, and explains observations about real data being learned faster than random data. Intriguingly our theory also reveals the existence of a learning algorithm that proveably out-performs neural network training through gradient descent. Finally, for transfer learning, our theory reveals that knowledge transfer depends sensitively, but computably, on the SNRs and input feature alignments of pairs of tasks.

## 1 INTRODUCTION

Many deep learning practitioners closely monitor both training and test errors, hoping to achieve both a small training error and a small generalization error, or gap between testing and training errors. Training is usually stopped early, before overfitting sets in and increases the test error. This procedure often results in large networks that generalize well on structured tasks, raising an important generalization puzzle (Zhang et al., 2016): many existing theories that upper bound generalization error (Bartlett & Mendelson, 2002; Neyshabur et al., 2015; Dziugaite & Roy, 2017; Golowich et al., 2017; Neyshabur et al., 2017; Bartlett et al., 2017; Arora et al., 2018, e.g) in terms of various measures of network complexity yield very loose bounds. Therefore they cannot explain the impressive generalization capabilities of deep nets.

In the absence of any such tight and computable theory of deep network generalization error, we develop an analytic theory of generalization error for deep linear networks. Such networks exhibit highly nonlinear learning dynamics (Saxe et al., 2013a;b) including many prominent phenomena like learning plateaus, saddle points, and sudden drops in training error. Moreover, theory developed for the learning dynamics of deep linear networks directly inspired better initialization schemes for nonlinear networks (Schoenholz et al., 2016; Pennington et al., 2017; 2018). Here we show that deep linear networks also provide a good theoretical model for generalization dynamics. In particular we develop an analytic theory for both the training and test error of a deep linear network as a function of training time, number of training examples, network architecture, initialization, and

task structure and SNR. Our theory matches simulations and reveals that deep networks with small weight initialization learn the most important aspects of a task first. Thus the optimal test error at the early stopping time depends largely on task structure and SNR, and not on network architecture, as long as the architecture is expressive enough to attain small training error. Thus our exact analysis of generalization dynamics reveals the important lesson that any theory that seeks to upper bound generalization error based only on network architecture, and not on task structure, is likely to yield exceedingly loose upper bounds. Intriguingly our theory also reveals a non-gradient-descent learning algorithm that provably out-performs neural network training through gradient descent.

We also apply our theory to multi-task learning, which enables knowledge transfer from one task to another, thereby further lowering generalization error (Dong et al., 2015; Rusu et al., 2015; Luong et al., 2016, e.g.). Moreover, knowledge transfer across tasks may be key to human generalization capabilities (Hansen et al., 2017; Lampinen et al., 2017). We provide an analytic theory for how much knowledge is transferred between pairs of tasks, and we find that it displays a sensitive but computable dependence on the relationship between pairs of tasks, in particular, their SNRs and feature space alignments.

We note that a related prior work (Advani & Saxe, 2017) studied generalization in shallow and deep linear networks, but that work was limited to networks with a single output, thereby precluding the possibility of addressing the issue of transfer learning. Moreover, analyzing networks with a single output also precludes the possibility of addressing interesting tasks that require higher dimensional outputs, for example in language (Dong et al., 2015, e.g.), generative models (Goodfellow et al., 2014, e.g), and reinforcement learning (Mnih et al., 2015; Silver et al., 2016, e.g).

## 2 THEORETICAL FRAMEWORK

We work in a student-teacher scenario in which we consider an ensemble of low rank, noisy teacher networks that generate training data for a potentially more complex student network, and define the training and test errors whose dynamics we wish to understand.

### 2.1 AN ENSEMBLE OF LOW-RANK NOISY TEACHERS

We first consider an ensemble of 3-layer linear teacher networks with $\overline{N}_i$ units in layer $i$, and weight matrices $\overline{\mathbf{W}}^{21} \in \mathbb{R}^{\overline{N}_2 \times \overline{N}_1}$ and $\overline{\mathbf{W}}^{32} \in \mathbb{R}^{\overline{N}_3 \times \overline{N}_2}$ between the input to hidden, and hidden to output layers, respectively. The teacher network thus computes the composite map $\overline{\mathbf{y}} = \overline{\mathbf{W}}\mathbf{x}$, where $\overline{\mathbf{W}} \equiv \overline{\mathbf{W}}^{32}\overline{\mathbf{W}}^{21}$. Of critical importance is the singular value decomposition (SVD) of $\overline{\mathbf{W}}$:

$$\overline{\mathbf{W}} = \overline{\mathbf{U}}\,\overline{\mathbf{S}}\,\overline{\mathbf{V}}^T = \sum_{\alpha=1}^{\overline{N}_2} \overline{s}^\alpha \overline{\mathbf{u}}^\alpha \overline{\mathbf{v}}^{\alpha T}, \tag{1}$$

Where $\overline{\mathbf{U}} \in \mathbb{R}^{\overline{N}_3 \times \overline{N}_2}$ and $\overline{\mathbf{V}} \in \mathbb{R}^{\overline{N}_1 \times \overline{N}_2}$ are both matrices with orthonormal columns and $\overline{\mathbf{S}}$ is an $\overline{N}_2 \times \overline{N}_2$ diagonal matrix. We construct a random teacher by picking $\overline{\mathbf{U}}$ and $\overline{\mathbf{V}}$ to be random matrices with orthonormal columns and choosing $O(1)$ values for the diagonal elements of $\overline{\mathbf{S}}$. We work in the limit $\overline{N}_1, \overline{N}_3 \to \infty$ with an $O(1)$ aspect ratio $\mathcal{A} = \overline{N}_3/\overline{N}_1 \in (0, 1]$ so that the teacher has fewer outputs than inputs. Also, we hold $\overline{N}_2 \sim O(1)$, so the teacher has a low, finite rank, and we study generalization performance as a function of the $\overline{N}_2$ teacher singular values.

We further assume the teacher generates noisy outputs from a set of $\overline{N}_1$ orthonormal inputs:

$$\hat{\mathbf{y}}^\mu = \overline{\mathbf{W}}\hat{\mathbf{x}}^\mu + \mathbf{z}^\mu \qquad \text{for} \quad \mu = 1, \ldots, \overline{\mathbf{N_1}}. \tag{2}$$

This training set yields important second-order training statistics that will guide student learning:

$$\mathbf{\Sigma}^{11} \equiv \sum_{\mu=1}^{\overline{N}_1} \hat{\mathbf{x}}^\mu \hat{\mathbf{x}}^{\mu T} = \mathbf{I}, \qquad \mathbf{\Sigma}^{31} \equiv \sum_{\mu=1}^{\overline{N}_1} \hat{\mathbf{y}}^\mu \hat{\mathbf{x}}^{\mu T} = \overline{\mathbf{W}} + \mathbf{Z}\hat{\mathbf{X}}^T. \tag{3}$$

Here the input covariance $\mathbf{\Sigma}^{11}$ is assumed to be white (a common pre-processing step), the input-output covariance $\mathbf{\Sigma}^{31}$ is simplified using (2), and $\mathbf{Z} \in \mathbb{R}^{\overline{N}_3 \times \overline{N}_1}$ is the noise matrix, whose $\mu$'th

column is $\mathbf{z}^\mu$. Its matrix elements $z_i^\mu$ are drawn iid. from a Gaussian with zero mean and variance $\sigma_z^2/\overline{N}_1$. The noise scaling is chosen so the singular values of the teacher $\overline{\mathbf{W}}$ and the noise $\mathbf{Z}$ are both $O(1)$, leading to non-trivial generalization effects. As generalization performance will depend on the *ratio* of teacher singular values to the noise variance parameter $\sigma_z^2$, we simply set $\sigma_z = 1$ in the following. Thus we can think of teacher singular values as signal to noise ratios (SNRs).

Finally, we note that while we focus for ease of exposition in the main paper on the case of one hidden layer networks and a full orthonormal basis of $P = \overline{N}_1$ training inputs in the main paper, neither of these assumptions are essential to our theory. Indeed in Section 3.4 and App. A we extend our theory to networks of arbitrary depth, and in App. G we extend our theory to the case of white inputs with $P \neq \overline{N}_1$, obtaining a good match between theory and experiment in both cases.

## 2.2  STUDENT TRAINING AND TEST ERROR

Now consider a student network with $N_i$ units in each layer. We assume the first and last layers match the teacher (i.e. $N_1 = \overline{N}_1$ and $N_3 = \overline{N}_3$) but $N_2 \geq \overline{N}_2$, allowing the student to have more hidden units than the teacher. We also consider deeper students (see below and App. A). Now consider any student whose input-output map is given by $\mathbf{y} = \mathbf{W}^{32}\mathbf{W}^{21} \equiv \mathbf{W}\mathbf{x}$. Its training error on the teacher dataset in (2) and its test error over a distribution of new inputs are given by

$$\varepsilon_{\text{train}} \equiv \frac{\sum_{\mu=1}^{\overline{N}_1} ||\mathbf{W}\hat{\mathbf{x}}^\mu - \hat{\mathbf{y}}^\mu||_2^2}{\sum_{\mu=1}^{\overline{N}_1} ||\hat{\mathbf{y}}^\mu||_2^2}, \qquad \varepsilon_{\text{test}} \equiv \frac{\left\langle ||\mathbf{W}\overline{\mathbf{x}} - \overline{\mathbf{y}}||_2^2 \right\rangle}{\left\langle ||\overline{\mathbf{y}}||_2^2 \right\rangle}, \qquad (4)$$

respectively. Here $\hat{\mathbf{x}}^\mu$ and $\hat{\mathbf{y}}^\mu$ are the noisy training set inputs and outputs in (2), whereas $\overline{\mathbf{x}}$ denotes a random test input drawn from zero mean Gaussian with identity covariance, $\overline{\mathbf{y}}^\mu = \overline{\mathbf{W}}\overline{\mathbf{x}}^\mu$ is noise free teacher output, and $\langle \cdot \rangle$ denotes an average w.r.t the distribution of the test input $\overline{\mathbf{x}}$. Due to the orthonormality of the training and isotropy of the test inputs, both $\varepsilon_{\text{train}}$ and $\varepsilon_{\text{test}}$ can be expressed as

$$\varepsilon_{\text{train}} = \frac{\operatorname{Tr}\mathbf{W}^T\mathbf{W} - 2\operatorname{Tr}\mathbf{W}^T\mathbf{\Sigma}^{31} + \operatorname{Tr}\mathbf{\Sigma}^{31\,T}\mathbf{\Sigma}^{31}}{\operatorname{Tr}\mathbf{\Sigma}^{31\,T}\mathbf{\Sigma}^{31}}, \ \ \varepsilon_{\text{test}} = \frac{\operatorname{Tr}\mathbf{W}^T\mathbf{W} - 2\operatorname{Tr}\mathbf{W}^T\overline{\mathbf{W}} + \operatorname{Tr}\overline{\mathbf{W}}^T\overline{\mathbf{W}}}{\operatorname{Tr}\overline{\mathbf{W}}^T\overline{\mathbf{W}}}. \tag{5}$$

Both $\varepsilon_{\text{train}}$ and $\varepsilon_{\text{test}}$ can be further expressed in terms of the student, training data and teacher SVDs, which we denote by $\mathbf{W} = \mathbf{U}\mathbf{S}\mathbf{V}^T$, $\mathbf{\Sigma}^{31} = \hat{\mathbf{U}}\hat{\mathbf{S}}\hat{\mathbf{V}}^T$, and $\overline{\mathbf{W}} = \overline{\mathbf{U}}\,\overline{\mathbf{S}}\,\overline{\mathbf{V}}^T$ respectively. Specifically,

$$\varepsilon_{\text{train}} = \left[\sum_{\beta=1}^{\overline{N}_3} \hat{s}_\beta^2\right]^{-1} \left[\sum_{\alpha=1}^{N_2} s_\alpha^2 + \sum_{\beta=1}^{\overline{N}_3} \hat{s}_\beta^2 - 2\sum_{\alpha=1}^{N_2}\sum_{\beta=1}^{\overline{N}_3} s_\alpha \hat{s}_\beta \left(\mathbf{u}^\alpha \cdot \hat{\mathbf{u}}^\beta\right)\left(\mathbf{v}^\alpha \cdot \hat{\mathbf{v}}^\beta\right)\right], \tag{6}$$

$$\varepsilon_{\text{test}} = \left[\sum_{\beta=1}^{\overline{N}_2} \overline{s}_\beta^2\right]^{-1} \left[\sum_{\alpha=1}^{N_2} s_\alpha^2 + \sum_{\beta=1}^{\overline{N}_2} \overline{s}_\beta^2 - 2\sum_{\alpha=1}^{N_2}\sum_{\beta=1}^{\overline{N}_2} s_\alpha \overline{s}_\beta \left(\mathbf{u}^\alpha \cdot \overline{\mathbf{u}}^\beta\right)\left(\mathbf{v}^\alpha \cdot \overline{\mathbf{v}}^\beta\right)\right]. \tag{7}$$

Thus as the student learns, its training and test error dynamics depends on the alignment of the time-evolving student singular modes $\{s^\alpha, \mathbf{u}^\alpha, \mathbf{v}^\alpha\}$ with the fixed training data $\{\hat{s}^\alpha, \hat{\mathbf{u}}^\alpha, \hat{\mathbf{v}}^\alpha\}$ and teacher $\{\overline{s}^\alpha, \overline{\mathbf{u}}^\alpha, \overline{\mathbf{v}}^\alpha\}$ singular modes respectively.

## 3  SINGLE TASK GENERALIZATION DYNAMICS: THEORY AND EXPERIMENT

Here we derive and numerically test analytic formulas for both the training and test errors of a student network as it learns from training data generated from a teacher network. We explore the dependence of these quantitites on the student network size, student initialization, teacher SNR, and training time.

### 3.1  STUDENT TRAINING DYNAMICS AND TRAINING-ALIGNED (TA) NETWORKS

We assume the student weights undergo batch gradient descent with learning rate $\lambda$ on the training error $\sum_\mu ||\hat{\mathbf{y}}^\mu - \mathbf{W}^{32}\mathbf{W}^{21}\hat{\mathbf{x}}^\mu||_2^2$, which for small $\lambda$ is well approximated by the differential equations:

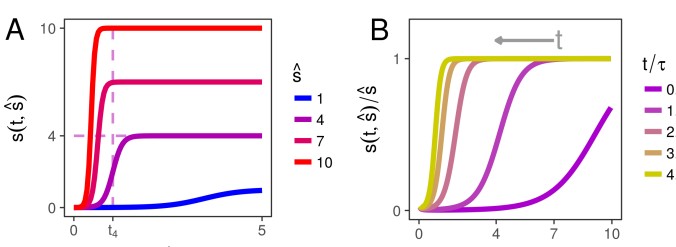

Figure 1: Learning dynamics as a function of singular dimension strength. (a) shows how modes of different singular value are learned, (b) shows that there is a wave of learning that picks up singular dimensions with smaller and smaller singular values as $t \to \infty$.

$$\tau \frac{d}{dt}\mathbf{W^{21}} = \mathbf{W^{32}}^T \left(\mathbf{\Sigma}^{31} - \mathbf{W^{32}W^{21}\Sigma^{11}}\right), \qquad \tau \frac{d}{dt}\mathbf{W^{32}} = \left(\mathbf{\Sigma}^{31} - \mathbf{W^{32}W^{21}\Sigma^{11}}\right)\mathbf{W^{21}}^T,$$
(8)

(where $\tau \equiv 1/\lambda$), which must be solved from an initial set of student weights at time $t = 0$ (Saxe et al., 2013a). We consider two classes of student initializations. The first initialization corresponds to a *random student* where the weights $\mathbf{W^{21}}$ and $\mathbf{W^{32}}$ are chosen such that the composite map $\mathbf{W} = \mathbf{W^{32}W^{21}}$ has an SVD $\mathbf{W} = \epsilon \mathbf{UV^T}$, where $\mathbf{U}$ and $\mathbf{V}$ are random singular vector matrices and all student singular values are $\epsilon$. As such a random student learns, the composite map undergoes a time dependent evolution $\mathbf{W}(t) = \mathbf{U}(t)\mathbf{S}(t)\mathbf{V}(t)^T = \sum_{\alpha=1}^{N_2} s_\alpha(t)\mathbf{u}^\alpha(t)\mathbf{v}^\alpha(t)^T$. For white inputs, as $t \to \infty$, $\mathbf{W} \to \mathbf{\Sigma}^{31}$, and so the time-dependent student singular modes $\{s^\alpha(t), \mathbf{u}^\alpha(\mathbf{t}), \mathbf{v}^{(}\mathbf{t})\}$ converge to the training data singular modes $\{\hat{s}^\alpha, \hat{\mathbf{u}}^\alpha, \hat{\mathbf{v}}^\alpha\}$. However, the explicit dynamics of the student singular modes can be difficult to obtain analytically from random initial conditions.

Thus we also consider a special class of *training aligned* (TA) initial conditions in which $\mathbf{W^{21}}$ and $\mathbf{W^{32}}$ are chosen such that the composite map $\mathbf{W} = \mathbf{W^{32}W^{21}}$ has an SVD $\mathbf{W} = \epsilon \hat{\mathbf{U}}\hat{\mathbf{V}}^T$. That is, the TA network (henceforth referred to simply as the TA) has the same singular vectors as the training data covariance $\mathbf{\Sigma}^{31}$, but has all singular values equal to $\epsilon$. As shown in (Saxe et al., 2013a), as the TA learns according to (8), the singular vectors of its composite map $\mathbf{W}$ remain unchanged, while the singular values evolve as $s^\alpha(t) = s(t, \hat{s}^\alpha)$, where the learning curve function $s(t, \hat{s})$ as well as its functional inverse $t(s, \hat{s})$ is given by

$$s(t, \hat{s}) = \frac{\hat{s} e^{2\hat{s}t/\tau}}{e^{2\hat{s}t/\tau} - 1 + \hat{s}/\epsilon}, \qquad t(s, \hat{s}) = \frac{\tau}{2\hat{s}} \ln \frac{\hat{s}/\epsilon - 1}{\hat{s}/s - 1}.$$
(9)

Here the function $s(t, \hat{s})$ describes analytically how each training set singular value $\hat{s}$ drives the dynamics of the corresponding TA singular value $s$, and for notational simplicity, we have suppressed the dependence of $s(t, \hat{s})$ on $\tau$ and the initial condition $\epsilon$. As shown in Fig. 1A, for each $\hat{s}$, $s(t, \hat{s})$ is a sigmoidal learning curve that undergoes a sharp transition around time $t/\tau = \frac{1}{2\hat{s}} \ln (\hat{s}/\epsilon - 1)$, at which it rises from its small initial value of $\epsilon$ at $t = 0$ to its asymptotic value of $\hat{s}$ as $t/\tau \to \infty$. Alternatively, we can plot $s(t, \hat{s})/\hat{s}$ as a function of $\hat{s}$ for different training times $t/\tau$, as in Fig. 1B. This shows that TA learning corresponds to a *singular mode detection wave* which progressively sweeps from large to small singular values. At any given training time $t$, training data modes with singular values $\hat{s} > t/\tau$ have been learned, while those with singular values $\hat{s} < t/\tau$ have not.

While the TA is more sophisticated than the random student, since it already knows the singular vectors of the training data before learning, we will see that the analytic solution for the TA learning dynamics provides a good approximation to the student learning dynamics, not only for the training error, as shown in (Saxe et al., 2013a), but also for the generalization error as shown below.

The results in this section assume a single hidden layer, but Saxe et al. (2013a) derived $t(s, \hat{s})$ for networks of arbitrary depth and we apply our theory to some deeper networks. The general differential equation and derivations for deeper networks can be found in Appendix A.

## 3.2 How the teacher is buried in the training data: a random matrix analysis

In the previous section, we reviewed an exact analytic solution for the composite map of a TA network, namely that its singular modes are related to those of the training data through the relation

$$s_\alpha(t) = s(t, \hat{s}_\alpha), \qquad \mathbf{u}^\alpha(t) = \hat{\mathbf{u}}^\alpha, \qquad \mathbf{v}^\alpha(t) = \hat{\mathbf{v}}^\alpha.$$
(10)

However, computation of the generalization error through (5) then requires understanding how the teacher singular modes of $\overline{\mathbf{W}}$ are buried within the noisy training data singular modes of $\mathbf{\Sigma}^{31}$ through

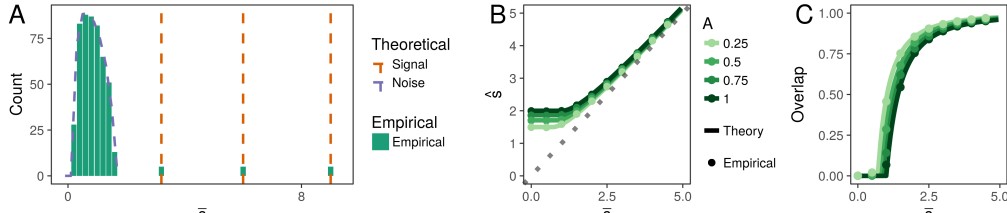

Figure 2: The teacher's signal through the noise. Theoretical vs. empirical (a) histogram of singular values of noisy teacher $\hat{s}$. (b) $\hat{s}$ as a function of $\overline{s}$. (c) alignment of noisy teacher and noiseless teacher singular vectors as a function of $\overline{s}$. ($\overline{N_1} = \overline{N_3} = 100$.)

the relation (3). Since the input matrix $\hat{\mathbf{X}}$ is orthonormal, $\mathbf{\Sigma^{31}}$ is simply a perturbation of the low rank teacher $\overline{\mathbf{W}}$ by a high dimensional noise matrix $\mathbf{Z}$. The relation between the singular modes of a low rank matrix and its noise perturbed version has been studied extensively in Benaych-Georges & Nadakuditi (2012), in the high dimensional limit we are working in, namely $\overline{N_1}, \overline{N_3} \to \infty$ with the aspect ratio $\mathcal{A} = \overline{N_3}/\overline{N_1} \in (0, 1]$, and $\overline{N_2} \sim O(1)$.

In this limit, the top $\overline{N_2}$ singular values and vectors of $\mathbf{\Sigma^{31}}$ converge to $\hat{s}(\overline{s}_\alpha)$, where the transfer function from a teacher singular value $\overline{s}$ to a training data singular value $\hat{s}$ is given by the function

$$\hat{s}(\overline{s}) = \begin{cases} (\overline{s})^{-1}\sqrt{(1+\overline{s}^2)(\mathcal{A}+\overline{s}^2)} & \text{if } \overline{s} > \mathcal{A}^{1/4} \\ 1 + \sqrt{\mathcal{A}} & \text{otherwise.} \end{cases} \tag{11}$$

The associated top $\overline{N_2}$ singular vectors of $\mathbf{\Sigma^{31}}$ can also acquire a nontrivial overlap with the $\overline{N_2}$ modes of the teacher through the relation $|\hat{\mathbf{u}}^\alpha \cdot \overline{\mathbf{u}}^\alpha| |\hat{\mathbf{v}}^\alpha \cdot \overline{\mathbf{v}}^\alpha| = \mathcal{O}(\overline{s}_\alpha)$, where the singular vector overlap function is given by

$$\mathcal{O}(\overline{s}) = \begin{cases} \left[1 - \frac{\mathcal{A}(1+\overline{s}^2)}{\overline{s}^2(\mathcal{A}+\overline{s}^2)}\right]^{1/2} \left[1 - \frac{(\mathcal{A}+\overline{s}^2)}{\overline{s}^2(1+\overline{s}^2)}\right]^{1/2} & \text{if } \overline{s} > \mathcal{A}^{1/4} \\ 0 & \text{otherwise} \end{cases} \tag{12}$$

The rest of the $N_3 - \overline{N_2}$ singular vectors of $\mathbf{\Sigma^{31}}$ are orthogonal to the top $\overline{N_2}$ ones, and their singular values are distributed according to the the Marchenko-Pastur (MP) distribution:

$$P(\hat{s}) = \begin{cases} \frac{\sqrt{4\mathcal{A}-(\hat{s}^2-(1+\mathcal{A}))^2}}{\pi\mathcal{A}\hat{s}} & \hat{s} \in [1-\sqrt{\mathcal{A}}, 1+\sqrt{\mathcal{A}}] \\ 0 & \text{otherwise.} \end{cases} \tag{13}$$

Overall, these equations describe a singular vector *phase transition* in the training data, as illustrated in Fig. 2BC. For example in the case of no teacher, the training data is simply noise and the singular values of $\mathbf{\Sigma^{31}}$ are distributed as an MP sea spread between $1 \pm \sqrt{\mathcal{A}}$. When one adds a teacher, how each teacher singular mode is imprinted on the training data depends crucially on the teacher singular value $\overline{s}$, and the nature of this imprinting undergoes a phase transition at $\overline{s} = \mathcal{A}^{1/4}$. For $\overline{s} \leq \mathcal{A}^{1/4}$, the teacher mode SNR is too low and this mode is not imprinted in the noisy training data; the associated training data singular value $\hat{s}$ remains at the edge of the MP sea at $1 + \sqrt{\mathcal{A}}$, and the overlap $\mathcal{O}(\overline{s})$ between training and teacher singular vectors remains zero.

However, when $\overline{s} > \mathcal{A}^{1/4}$, this teacher mode is imprinted in the training data; there is an associated training data singular value $\hat{s}$ that pops out of the MP sea (Fig. 2AB). However, the training data singular value emerges at a position $\hat{s} > \overline{s}$ that is *inflated* by the noise, though the inflation effect decreases at larger $\overline{s}$, with the ratio $\hat{s}/\overline{s}$ approaching the unity line as $\overline{s}$ becomes large (Fig. 2B). Similarly, the corresponding training data singular vectors acquire a non-trivial overlap with the teacher singular vectors when $\overline{s} > \mathcal{A}^{1/4}$, and the alignment approaches unity as $\overline{s}$ increases (Fig. 2C).

### 3.3 PUTTING IT ALL TOGETHER: AN ANALYTIC THEORY OF GENERALIZATION DYNAMICS

Based on an analytic understanding of how the singular mode structure $\{\overline{s}^\alpha, \overline{\mathbf{u}}^\alpha, \overline{\mathbf{v}}^\alpha\}$ of the teacher $\overline{\mathbf{W}}$ is imprinted in the modes $\{\hat{s}^\alpha, \hat{\mathbf{u}}^\alpha, \hat{\mathbf{v}}^\alpha\}$ of the training data covariance $\mathbf{\Sigma^{31}}$ through (11), (12) and (13), and in turn how this training data singular structure drives the time evolving singular modes of a

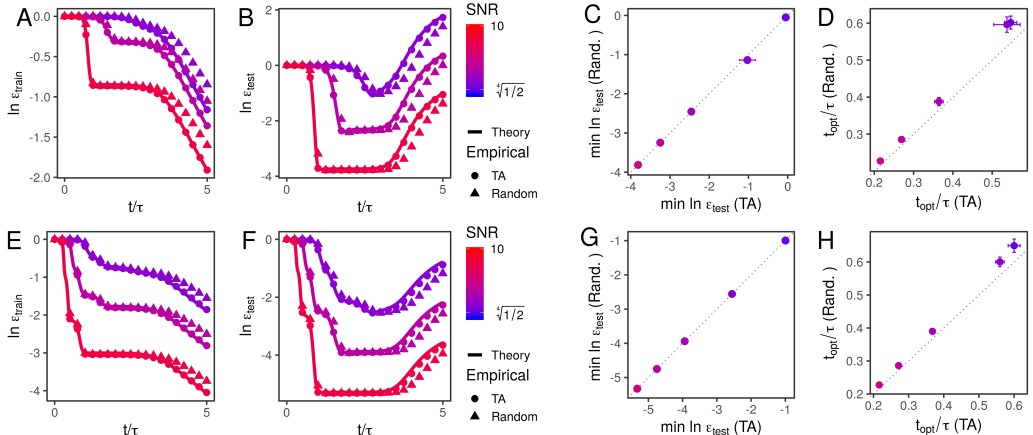

Figure 3: Match between theory and experiment for rank 1 (row 1, a-d) and rank 3 (row 2, e-h) teachers with single-hidden-layer students: (a-b, e-f) log train and test error, respectively, showing very close match between theory and experiment for TA, and close match for the random student. (c,g) comparing TA and randomly initialized students minimum generalization errors, showing almost perfect match. (d,h) comparing TA and randomly initialized students optimal stopping times, showing small lag due to alignment. ($N_1 = 100$, $N_2 = 50$, $N_3 = 50$.)

TA network $\{s^\alpha(t), \hat{\mathbf{u}}^\alpha, \hat{\mathbf{v}}^\alpha\}$ of through (9), we can now derive analytic expressions for $\varepsilon_{\text{train}}$ and $\varepsilon_{\text{test}}$ in (6) and (7), for a TA network. We will also show that these learning curves closely approximate those of a random student with time-evolving singular vectors $\{\mathbf{u}^\alpha(t), \mathbf{v}^\alpha(t)\}$, and match on several key aspects. First, inserting the TA dynamics in (10) into $\varepsilon_{\text{train}}$ in (6), we obtain

$$\varepsilon_{\text{train}}(t) = \left[\sum_{\alpha=1}^{\overline{N}_3} \hat{s}_\alpha^2\right]^{-1}\left[(N_3 - N_2)\langle \hat{s}^2\rangle_{\mathcal{R}_{out}} + (N_2 - \overline{N}_2)\langle(s(\hat{s}, t) - \hat{s})^2\rangle_{\mathcal{R}_{in}} + \sum_{\alpha=1}^{\overline{N}_2}[s_\alpha(t) - \hat{s}_\alpha]^2\right]$$

(14)

Here, $s_\alpha(t) = s(\hat{s}_\alpha, t)$ as defined in (9) are the TA singular values, and $\hat{s}_\alpha = \hat{s}(\overline{s}_\alpha)$ as defined in (11) are the training data singular values associated with the teacher singular values $\overline{s}_\alpha$. Also $\langle\cdot\rangle_{\mathcal{R}}$ denotes an average with respect to the MP distribution in (13) over a region $\mathcal{R}$. Two distinct regions contribute to training error. First $\mathcal{R}_{in}$ contains those top $N_2 - \overline{N}_2$ training data singular values that do not correspond to the $\overline{N}_2$ singular values of the teacher but will be learned by a rank $N_2$ student. Second, $\mathcal{R}_{out}$ corresponds to the remaining $N_3 - N_2$ lowest training data singular values that cannot be learned by a rank $N_2$ student. In terms of the MP distribution, $\mathcal{R}_{out} = [1 - \sqrt{\mathcal{A}}, f]$ and $\mathcal{R}_{in} = [f, 1 + \sqrt{\mathcal{A}}]$, where $f$ is the point at which the MP density has $1 - N_2/N_3$ of its mass to the left and $N_2/N_3$ of its mass to the right. In the simple case of a full rank student, $f = 1 - \sqrt{\mathcal{A}}$, and one need only integrate over $\mathcal{R}_{in}$ which is the entire range. Equation (14) for $\varepsilon_{\text{train}}$ makes it manifest that it will go to zero for a full rank student as its singular values approach those of the training data.

Of course the test error can behave very differently. Inserting the TA training dynamics in (10) into $\varepsilon_{\text{test}}$ in (7), and using (11), (12) and (13) to relate training data to the teacher, we find

$$\varepsilon_{\text{test}}(t) = \left[\sum_{\alpha=1}^{\overline{N}_2} \overline{s}_\alpha^2\right]^{-1}\left[(N_2 - \overline{N}_2)\langle s(\hat{s}, t)^2\rangle_{\mathcal{R}_{in}} + \sum_{\alpha=1}^{\overline{N}_2}\left[(s_\alpha(t) - \overline{s}_\alpha)^2 + 2s_\alpha(t)\overline{s}_\alpha(1 - \mathcal{O}(\overline{s}_\alpha))\right]\right]$$

(15)

Together (14) and (15) constitute a complete theory of generalization dynamics in terms of the structure of the data distribution (i.e. the teacher rank $\overline{N}_2$, teacher SNRs $\{\overline{s}_\alpha\}$, and the teacher aspect ratio $\mathcal{A} = \overline{N}_3/\overline{N}_1$), the architectural complexity of the student (i.e. its rank $N_2$, its number of layers $N_l$, and the norm $\epsilon$ of its initialization), and the training time $t$. They yield considerable insight into the dynamics of good generalization early in learning and overfitting later, as we show below.

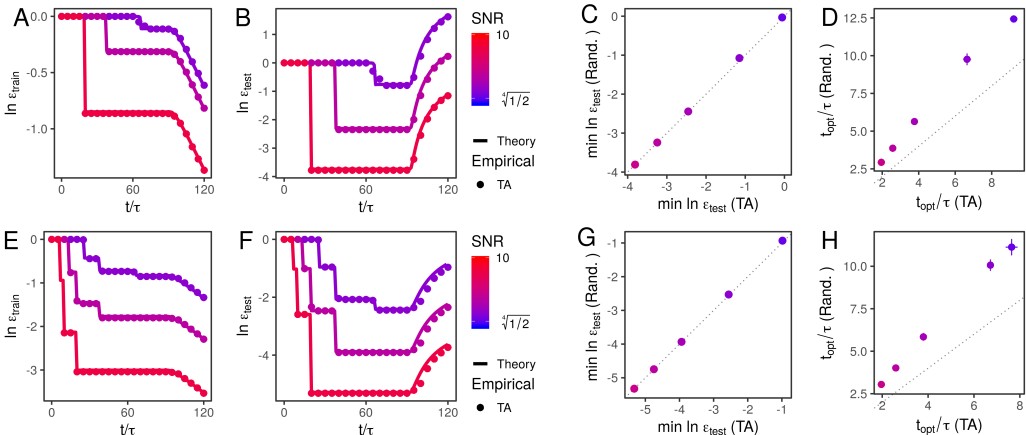

Figure 4: Our theory applies to deeper networks: match between theory and simulation for rank 1 (row 1, a-d) and rank 3 (row 2, e-h) teachers with $n_l = 5$ students: (a-b, e-f) log train and test error, respectively, showing very close match between theory and experiment for TA. (c,g) comparing TA and randomly initialized students minimum generalization errors, showing almost perfect match. (d,h) comparing TA and randomly initialized students optimal stopping times, showing large lag due to slower alignment in deeper networks. ($N_1 = 100$, $N_2 = 50$, $N_3 = 50$.)

### 3.4 NUMERICAL TESTS OF THE THEORY OF NEURAL NETWORK GENERALIZATION DYNAMICS

Fig. 3 demonstrates an excellent match between the theory and simulations for the TA, and a close match for random students, for single-hidden-layer students and various teacher ranks $\overline{N}_2$. Intuitively, as time $t$ proceeds, learning corresponds to singular mode detection wave sweeping from large to small training data singular values (i.e. the wave in Fig. 1B sweeps across the training data spectrum in Fig 2A). Initially, strong singular values associated with large SNR teacher modes are learned and both $\varepsilon_{\text{train}}$ and $\varepsilon_{\text{test}}$ drop. Fig. 3A-D are for a rank 1 teacher, and so in Fig 3AB we see a single sharp drop early on, if the teacher SNR is sufficiently high. By contrast, with a rank 3 teacher in Fig. 3E-H, there are several early drops as the three modes are picked up. However, as time progresses, the singular mode detection wave penetrates the MP sea, and the student picks up noise structure in the data, so $\varepsilon_{\text{train}}$ drops but $\varepsilon_{\text{test}}$ rises, indicating the onset of overfitting.

The main difference between the random student and TA learning curves is that the random student learning is slightly delayed relative to the TA, especially late in training. This is understandable because the TA already knows the singular vectors of the training data, while the random student must learn them. Nevertheless, two of the most important aspects of learning, namely the optimal early stopping time $t_{\text{gradient}}^{\text{opt}} \equiv \text{argmin}_t \varepsilon_{\text{test}}(t)$ and the minimal test error achieved at this time $\varepsilon_{\text{gradient}}^{\text{opt}} \equiv \min_t \varepsilon_{\text{test}}(t)$, match well between TA and random student, as shown in Fig. 3CD. At low teacher SNRs, the student takes a little longer to learn than the TA, but their optimal test errors match.

Our theory can also be easily extended to describe the learning dynamics deeper networks. Saxe et al. (2013a) derived $t(s, \hat{s})$ for networks of arbitrary depth, so we only need to adjust this factor in our formulas, see App. A for details. In Fig. 4 we show that again there is an excellent match between TA networks and theory for student networks with $N_l = 5$ layers (i.e. 3 hidden layers). Randomly-initialized networks show a much longer alignment lag for deeper networks (see App. B for details), but the curves are qualitatively similar and optimal stopping errors match. We also demonstrate extensions of our theory to different numbers of training examples (App. G).

Importantly, many of the phenomena we observe in linear networks are qualitatively replicated in nonlinear networks (Fig. 5), suggesting that our theory may help guide understanding of the nonlinear case. In particular, features such as stage-like initial learning, followed by a plateau if SNR is high, and finally followed by overfitting, are replicated. However, there are some discrepancies, in particular nonlinear networks (especially deeper ones) begin overfitting earlier than linear networks. This is likely because a mode in a non-linear network can be co-opted by an orthogonal mode, while in a linear network it cannot. Thus noise modes are able to "stow away" on the strong signal modes once they are learned. However, overall learning patterns are similar, and we show below that

many interesting phenomena in nonlinear networks are understandable in the linear case, such as the (non-)effects of overparameterization, the dynamics of memorization, and the benefits of transfer.

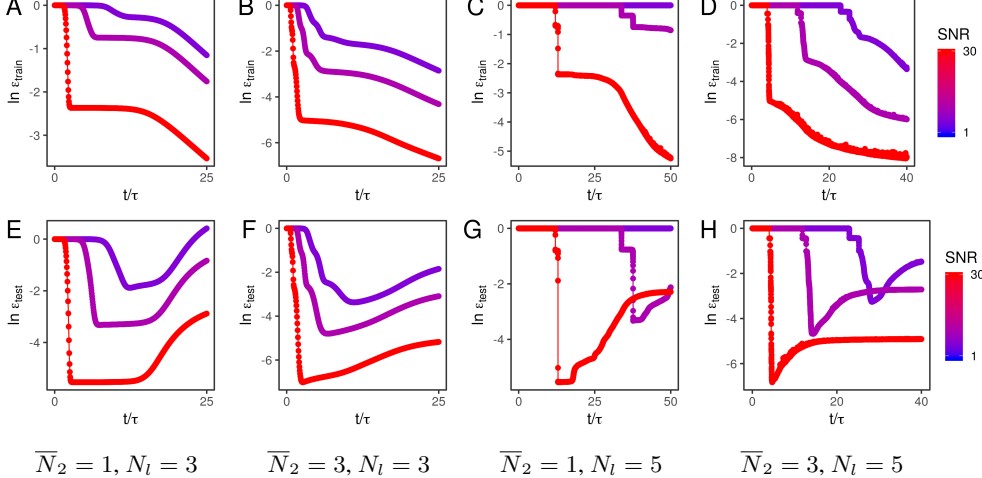

$$\overline{N}_2 = 1, N_l = 3 \qquad \overline{N}_2 = 3, N_l = 3 \qquad \overline{N}_2 = 1, N_l = 5 \qquad \overline{N}_2 = 3, N_l = 5$$

Figure 5: Train (first row, A-D) and test (second row, E-H) error for nonlinear networks (leaky relu at all hidden layers) with one hidden layer (first two columns) or three hidden layers (last two columns) trained on the tasks above, with a rank 1 teacher (first and third columns) or a rank 3 teacher (second and fourth columns). Note that many of the qualitative phenomena observed in linear networks, such as stage-like improvement in the errors, followed by a plateau, followed by overfitting, also appear in nonlinear networks. Compare the first column to Fig. 3AB, the second column to Fig. 3EF, the third to Fig. 4AB, and the fourth to Fig. 4EF. ($N_1 = 100, N_2 = 50, N_3 = 50$.)

## 3.5 RANDOMIZED DATA VS. REAL DATA: A LEARNING TIME PUZZLE

An intriguing observation that resurrected the generalization puzzle in deep learning was the observation by Zhang et al. (2016) that deep networks can memorize data with the labels randomly permuted. However, as Arpit et al. (2017) pointed out, the learning dynamics of training error for randomized labels can be slower than than for structured data. This phenomenon also arises in deep linear networks, and our theory yields an analytic explanation for why. We randomize data by choosing orthonormal inputs $\hat{\mathbf{x}}^\mu$ as in the structured case, but we choose the outputs $\hat{\mathbf{y}}^\mu$ to be i.i.d. Gaussian with zero mean and the same diagonal variance as the structured training data generated by the teacher. For structured data generated by a low rank teacher with singular values $\bar{s}_\alpha$, the diagonal output variance is given by $\sigma_r^2 = \frac{1}{N_3}\left[\sum_{i=\alpha}^{\overline{N}_2} \bar{s}_\alpha^2\right] + \frac{1}{N_1}\sigma_z^2$, where $\sigma_z$ is the noise variance, as before. Since there is no relation between input and output, $\mathbf{\Sigma}^{31}$ is now distributed as a MP distribution whose support is $[(\sigma_r(1 - \sqrt{\mathcal{A}}), \sigma_r(1 + \sqrt{\mathcal{A})}]$. Thus randomization essentially destroys the outlier signal singular values in $\mathbf{\Sigma}^{31}$ reflecting the teacher, and distributes them across all randomized data

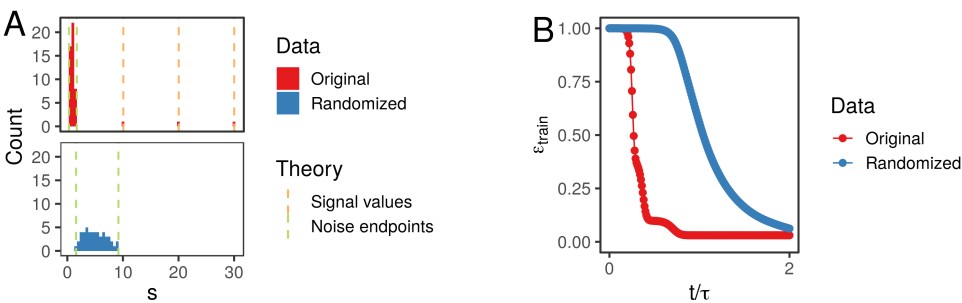

Figure 6: Learning randomized data: Comparing (a) singular value distributions and (b) learning curves for data with a signal vs. random data that preserves basic statistics (mean, variance). Randomizing the data dilutes the signal singular values, spreading their variance out over many modes, hence randomly labelled data is learned more slowly. ($N_1 = 100, N_2 = 50, N_3 = 50$.)

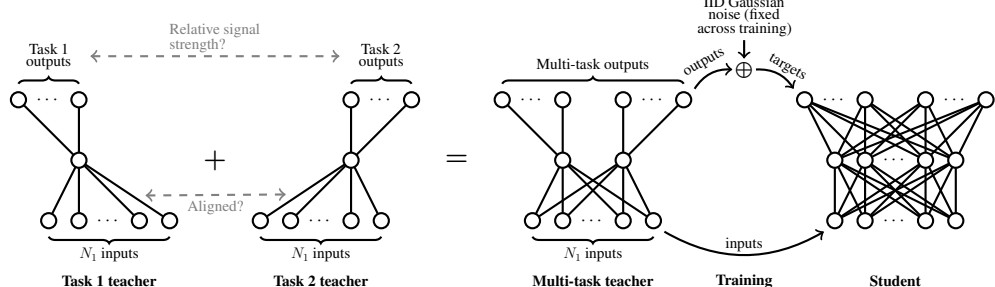

Figure 7: Transfer setting– If two different tasks are combined, how well students of the combined teacher perform on each task depends on the alignment and SNRs of the teachers.

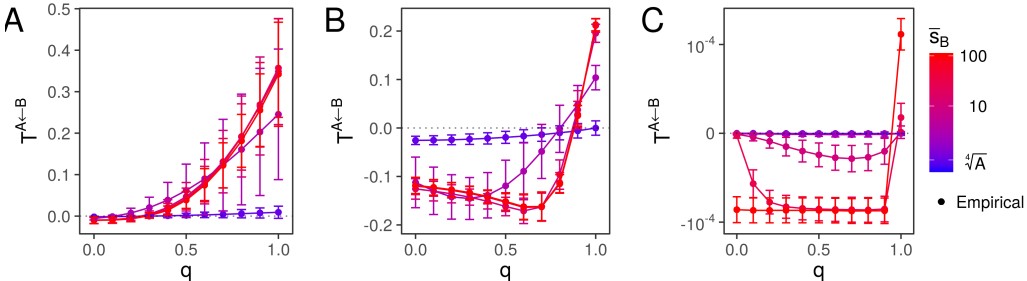

Figure 8: Transfer benefit $\mathcal{T}^{A \leftarrow B}(\overline{s}_A, \overline{s}_B, q)$ plotted at different values of $\overline{s}_A$. (a) $\overline{s}_A = 0.84 = \sqrt[4]{\mathcal{A}}$. Although this task is impossible to learn on its own, with support from another aligned task, especially one with high SNR, learning can occur. (b) $\overline{s}_A = 3$. Tasks with modest signals will face interference from poorly aligned tasks, but benefits from well aligned tasks. These effects are amplified by SNR. (c) $\overline{s}_A = 100$. Tasks with very strong signals will show little effect from other tasks (note y-axis scales), but any impact will be negative unless the tasks are very well aligned. ($N_1 = 100$ $\overline{N}_2^A = \overline{N}_2^B = 1$, $N_2 = 50$, $N_3 = 50$.)

modes, yielding this stretched MP distribution (compare 6A top and bottom). However, even on this stretched MP distribution, the right edge will be much smaller than the signal singular values, since the signal variance will be diluted by spreading it out over many more modes in the randomized data. Thus the randomized data will lead to slower initial training error drops relative to the structured data (Fig. 6B) since the singular mode detection wave encounters the first signal singular values in structured data earlier than it encounters the edge of the stretched MP sea in randomized data.

### 3.6 OUT-PERFORMING OPTIMAL EARLY STOPPING THROUGH A NON-GRADIENT ALGORITHM

For the case of a rank 1 teacher, it is straightforward to derive a good analytic approximation to the important quantities $\varepsilon_{\text{gradient}}^{\text{opt}}$ and $t_{\text{gradient}}^{\text{opt}}$. We assume the teacher SNR is beyond the phase transition point so its unique singular value $\overline{s}_1 > \mathcal{A}^{1/4}$, yielding a separation between the training data singular value $\hat{s}_1$ in (11) and the edge of the MP sea. In this scenario, optimal early stopping will occur at a time *before* the detection wave in Fig. 1B penetrates the MP sea, so to minimize test error, we can neglect the first term in (15). Then optimizing the second term yields the optimal student singular value $s_1 = \overline{s}_1 \mathcal{O}(\overline{s}_1)$. Inserting this value into (15) yields $\varepsilon_{\text{gradient}}^{\text{opt}} = 1 - \mathcal{O}(\overline{s}_1)^2$, and inserting it into (9) yields $t_{\text{gradient}}^{\text{opt}}$. Thus the optimal generalization error with a rank 1 teacher is very simply related to the alignment of the top training data singular vectors with the teacher singular vectors, and it decreases as this alignment increases. In App. E, we show this match in the rank 1 case.

With higher rank teachers, $\varepsilon_{\text{gradient}}^{\text{opt}}$ and $t_{\text{gradient}}^{\text{opt}}$ must negotiate a more complex trade-off between teacher modes with different SNRs. For example, as the singular mode detection wave passes the top training data singular value, $s_1(t) \rightarrow \hat{s}_1$ which is greater than the optimal $s_1 = \overline{s}_1 \mathcal{O}(\overline{s}_1)$ for mode 1. Thus as learning progresses, the student overfits on the first mode but learns lower modes. However, this neural generalization dynamics suggests a *superior non-gradient* training algorithm that simply

optimally sets each $s_\alpha$ to $\overline{s}_\alpha \mathcal{O}(\overline{s}_\alpha)$ in (15), yielding an optimal generalization error:

$$\varepsilon_{\text{non-gradient}}^{\text{opt}} = \left[ \sum_{\alpha=1}^{\overline{N}_2} \overline{s}_\alpha^2 \right]^{-1} \left[ \sum_{\alpha=1}^{\overline{N}_2} \overline{s}_\alpha^2 (1 - \mathcal{O}(\overline{s}_\alpha)^2) \right]. \tag{16}$$

Standard gradient descent learning cannot achieve this low generalization error because it cannot independently adjust all student singular values. A simple algorithm that achieves $\varepsilon_{\text{non-gradient}}^{\text{opt}}$ is as follows. From the training data covariance $\mathbf{\Sigma}^{31}$, extract the top singular values $\hat{s}_\alpha$ that pop-out of the MP sea, use the functional inverse of (11) to compute $\overline{s}_a(\hat{s}_\alpha)$, use (12) to compute the optimal $s_\alpha$, and then construct a matrix $\mathbf{W}$ with the same top singular vectors as $\mathbf{\Sigma}^{31}$, but with the outlier singular values shrunk from $\hat{s}_\alpha$ to $s_\alpha$ and the rest set to zero. This non-gradient singular value shrinkage algorithm provably outperforms neural network training with $\varepsilon_{\text{non-gradient}}^{\text{opt}} \le \varepsilon_{\text{gradient}}^{\text{opt}}$.

## 4    A THEORY FOR THE TRANSFER OF KNOWLEDGE ACROSS MULTIPLE TASKS

Consider two tasks $A$ and $B$, described by $\overline{N}_3$ by $\overline{N}_1$ teacher maps $\overline{\mathbf{W}}^A$ and $\overline{\mathbf{W}}^B$, of ranks $\overline{N}_2^A$ and $\overline{N}_2^B$, respectively. Now two student networks can learn from the two teacher networks separately, each achieving optimal early stopping test errors $\varepsilon_A^{\text{opt}}$ and $\varepsilon_B^{\text{opt}}$. Alternatively, one could construct a composite teacher (and student) that concatenates the hidden and output units, but shares the same input units (Fig. 7). The composite student and teacher each have two heads, one for each task, with $\overline{N}_3$ neurons per head. Optimal early stopping on each head of the student yields test errors $\varepsilon_{A \leftarrow B}^{\text{opt}}$ and $\varepsilon_{B \leftarrow A}^{\text{opt}}$. We define the *transfer benefit* that task B confers on task A to be $\mathcal{T}^{A \leftarrow B} \equiv \varepsilon_A^{\text{opt}} - \varepsilon_{A \leftarrow B}^{\text{opt}}$. A postive (negative) transfer benefit implies learning tasks A and B simultaneously yields a lower (higher) optimal test error on task A compared to just learning task A alone.

A foundational question is how the transfer benefit $\mathcal{T}^{A \leftarrow B}$ depends on the two tasks defined by the teachers $\overline{\mathbf{W}}^A$ and $\overline{\mathbf{W}}^B$. To answer this, consider the SVDs of each teacher alone: $\overline{\mathbf{W}}^A = \overline{\mathbf{U}}^A \overline{\mathbf{S}}^A \overline{\mathbf{V}}^{A^T}$ and $\overline{\mathbf{W}}^B = \overline{\mathbf{U}}^B \overline{\mathbf{S}}^B \overline{\mathbf{V}}^{B^T}$. From the above, we know that $\varepsilon_A^{\text{opt}}$ depends on $\overline{\mathbf{W}}^A$ only through $\overline{\mathbf{S}}^A$. In App. D we show that the transfer benefit depends on both $\overline{\mathbf{W}}^A$ and $\overline{\mathbf{W}}^B$ *only* through $\overline{\mathbf{S}}^A, \overline{\mathbf{S}}^B$, and the $\overline{N}_2^A$ by $\overline{N}_2^B$ similarity matrix $\overline{\mathbf{Q}} = \overline{\mathbf{V}}^{A^T} \overline{\mathbf{V}}^B$. If we think of the columns of each $\overline{\mathbf{V}}$ as spanning a low dimensional feature space in $\overline{N}_1$ dimensional input space that is important for each task, then $\overline{\mathbf{Q}}$ reflects the input feature subspace similarity matrix. Interestingly, the transfer benefit is independent of output singular vectors $\overline{\mathbf{U}}^A$ and $\overline{\mathbf{U}}^B$. What matters for knowledge transfer in this setting are the relevant input features, not how you must respond to them.

We describe the transfer benefit for the simple case of two rank one teachers. Then $\overline{\mathbf{S}}^A, \overline{\mathbf{S}}^B$, and $\overline{\mathbf{Q}}$ are simply scalars $s_A$, $s_B$ and $q$, and we explore the function $\mathcal{T}^{A \leftarrow B}(\overline{s}_A, \overline{s}_B, q)$ in Fig. 5ABC, which reveals several interesting features. First, knowledge can be transferred from a high SNR task to a low SNR task (Fig. 5A) and the degree of transfer increases with task alignment $q$. This can make it possible to capture signals from task $A$ which would otherwise sink into the MP sea by learning jointly with a related task, even if the tasks are only weakly aligned (Fig. 5A). However, if task $A$ already has a high SNR, task $B$ must be very well aligned to it for transfer to be beneficial – otherwise there will be interference. The degree of alignment required increases as the task $A$ SNR increases, but the quantity of benefit or interference decreases correspondingly (Fig. 5BC). In Appendix D we explain why our theory predicts these results. Furthermore, in Appendix F we demonstrate these phenomena are qualitatively recapitulated in *nonlinear* networks, which suggests that our theory may give insight into how to choose auxiliary tasks.

## 5    DISCUSSION

In summary, our analytic theory of generalization dynamics in deep linear networks reveals that many puzzling aspects of generalization in deep learning already arise in the simple linear setting, where the puzzles can be understood analytically. In particular, deep linear networks learn more important structure in data first, leading to generalization errors that depend on task structure much more than network size. Our theory explains why deep linear networks learn randomized data more slowly than

structured data, and provides a non-gradient based learning method that out-performs gradient descent learning in the linear case. Finally, we provide an analytic theory of how knowledge is transferred from one task to another, demonstrating that the degree of alignment of input features important for each task, but not how one must respond to these features, is critical for facilitating knowledge transfer. We think these analytic results provide useful insight into the similar generalization and transfer phenomena observed in the nonlinear case. Among other things, we hope our work will motivate and enable: (1) the search for tighter upper bounds on generalization error that take into account task structure; (2) the design of non gradient based training algorithms that outperform gradient-based learning; and (3) the theory-driven selection of auxiliary tasks that maximize knowledge transfer.

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

## A LEARNING DYNAMICS FOR DEEPER NETWORKS

In the main text, we described the dynamics of how a single-hidden-layer network converges toward the training data singular modes $\{\hat{s}^\alpha, \hat{\mathbf{u}}^\alpha, \hat{\mathbf{v}}^\alpha\}$, which were originally derived in Saxe et al. (2013a). There it was also proven that for a network with $N_l$ layers (i.e. $N_l - 2$ hidden layers), the strength of the mode obeys the differential equation:

$$\tau \frac{d}{dt} u = (N_l - 1) u^{2-2/(N_l-1)} (s - u)$$

This equation is separable and can be integrated for any integer number of layers. In particular, we consider the case of 5 layers (3 hidden), in which case:

$$t(s, \hat{s}) = \frac{\tau}{2} \left[ \frac{\tanh^{-1}\left(\sqrt{\frac{u}{\hat{s}}}\right)}{\hat{s}^{3/2}} - \frac{1}{\hat{s}\sqrt{u}} \right]_\epsilon^s$$

This expression cannot be analytically inverted to find $s(t, \hat{s})$, so we numerically invert it where necessary.

# B  ALIGNMENT LAG IN RANDOMLY INITIALIZED NETWORKS

As noted in the main text, the randomly-initialized networks behave quite similarly to the TA networks, except that the randomly-initialized networks show a lag due to the time it takes for the network's modes to align with the data modes. In fig. 9 we explore this lag by plotting the alignment of the modes and the increase in the singular value for several randomly initialized networks.

Notice that stronger modes align more quickly. Furthermore, the mode alignment is relatively independent – whether the teacher is rank 1 or rank 3, the alignment of the modes is similar for the mode of singular value 2. Most importantly, note how the deeper networks show substantially slower mode alignment, with alignment not completed until around when the singular value increases. This explains why deeper networks show a larger lag between randomly-initialized and TA networks – the alignment process is much slower for deeper networks.

# C  TRAIN AND TEST ERRORS AFTER A PROJECTION

In the case of transfer learning, or more generally when we want to evaluate a network's loss on a subset of its outputs, we need to use a slight generalization of the train and test error formulas given in the main text. Suppose we are interested in the train and test errors after applying a projection operator $\mathbf{P}$:

$$\varepsilon_{\text{train}} \equiv \frac{\sum_{\mu=1}^{\overline{N_1}} ||\mathbf{P}\mathbf{W}\hat{\mathbf{x}}^\mu - \mathbf{P}\hat{\mathbf{y}}^\mu||_2^2}{\sum_{\mu=1}^{\overline{N_1}} ||\mathbf{P}\hat{\mathbf{y}}^\mu||_2^2}, \qquad \varepsilon_{\text{test}} \equiv \frac{\sum_{\mu=1}^{\overline{N_1}} ||\mathbf{P}\mathbf{W}\overline{\mathbf{x}}^\mu - \mathbf{P}\overline{\mathbf{y}}^\mu||_2^2}{\sum_{\mu=1}^{\overline{N_1}} ||\mathbf{P}\overline{\mathbf{y}}^\mu||_2^2}, \qquad (17)$$

respectively. As in the main text, we can rexpress these as:

$$\varepsilon_{\text{train}} = \frac{\operatorname{Tr}\mathbf{W}^T\mathbf{P}^T\mathbf{P}\mathbf{W} - 2\operatorname{Tr}\mathbf{W}^T\mathbf{P}^T\mathbf{P}\mathbf{\Sigma}^{31} + \operatorname{Tr}\mathbf{\Sigma}^{31^T}\mathbf{P}^T\mathbf{P}\mathbf{\Sigma}^{31}}{\operatorname{Tr}\mathbf{\Sigma}^{31^T}\mathbf{P}^T\mathbf{P}\mathbf{\Sigma}^{31}}, \qquad (18)$$

$$\varepsilon_{\text{test}} = \frac{\operatorname{Tr}\mathbf{W}^T\mathbf{P}^T\mathbf{P}\mathbf{W} - 2\operatorname{Tr}\mathbf{W}^T\mathbf{P}^T\mathbf{P}\overline{\mathbf{W}} + \operatorname{Tr}\overline{\mathbf{W}}^T\mathbf{P}^T\mathbf{P}\overline{\mathbf{W}}}{\operatorname{Tr}\overline{\mathbf{W}}^T\mathbf{P}^T\mathbf{P}\overline{\mathbf{W}}}. \qquad (19)$$

Using the cyclic property of the trace, we can modify these to get:

$$\varepsilon_{\text{train}} = \frac{\operatorname{Tr}\mathbf{P}\mathbf{W}\mathbf{W}^T\mathbf{P}^T - 2\operatorname{Tr}\mathbf{P}\mathbf{\Sigma}^{31}\mathbf{W}^T\mathbf{P}^T + \operatorname{Tr}\mathbf{P}\mathbf{\Sigma}^{31}\mathbf{\Sigma}^{31^T}\mathbf{P}^T}{\operatorname{Tr}\mathbf{P}\mathbf{\Sigma}^{31}\mathbf{\Sigma}^{31^T}\mathbf{P}^T}, \qquad (20)$$

$$\varepsilon_{\text{test}} = \frac{\operatorname{Tr}\mathbf{P}\mathbf{W}\mathbf{W}^T\mathbf{P}^T - 2\operatorname{Tr}\mathbf{P}\overline{\mathbf{W}}\mathbf{W}^T\mathbf{P}^T + \operatorname{Tr}\mathbf{P}\overline{\mathbf{W}}\,\overline{\mathbf{W}}^T\mathbf{P}^T}{\operatorname{Tr}\mathbf{P}\overline{\mathbf{W}}\,\overline{\mathbf{W}}^T\mathbf{P}^T}. \qquad (21)$$

As before, we express these in terms of the student, training data and teacher SVDs, $\mathbf{W} = \mathbf{U}\mathbf{S}\mathbf{V}^T$, $\mathbf{\Sigma}^{31} = \hat{\mathbf{U}}\hat{\mathbf{S}}\hat{\mathbf{V}}^T$, and $\overline{\mathbf{W}} = \overline{\mathbf{U}}\,\overline{\mathbf{S}}\,\overline{\mathbf{V}}^T$ respectively. Specifically,

$$\varepsilon_{\text{train}} = \left[\sum_{\beta=1}^{\overline{N_3}} \hat{s}_\beta^2 ||\mathbf{P}\hat{\mathbf{u}}^\alpha||_2^2\right]^{-1}\left[\sum_{\alpha=1}^{N_2} s_\alpha^2 ||\mathbf{P}\mathbf{u}^\alpha||_2^2 + \sum_{\beta=1}^{\overline{N_3}} \hat{s}_\beta^2 ||\mathbf{P}\hat{\mathbf{u}}^\alpha||_2^2 - 2\sum_{\alpha=1}^{N_2}\sum_{\beta=1}^{\overline{N_3}} s_\alpha\hat{s}_\beta\left(\mathbf{P}\mathbf{u}^\alpha\cdot\mathbf{P}\hat{\mathbf{u}}^\beta\right)\left(\mathbf{v}^\alpha\cdot\hat{\mathbf{v}}^\beta\right)\right], \qquad (22)$$

$$\varepsilon_{\text{test}} = \left[\sum_{\beta=1}^{\overline{N_2}} \overline{s}_\beta^2 ||\mathbf{P}\overline{\mathbf{u}}^\alpha||_2^2\right]^{-1}\left[\sum_{\alpha=1}^{N_2} s_\alpha^2 ||\mathbf{P}\mathbf{u}^\alpha||_2^2 + \sum_{\beta=1}^{\overline{N_2}} \overline{s}_\beta^2 ||\mathbf{P}\overline{\mathbf{u}}^\alpha||_2^2 - 2\sum_{\alpha=1}^{N_2}\sum_{\beta=1}^{\overline{N_2}} s_\alpha\overline{s}_\beta\left(\mathbf{P}\mathbf{u}^\alpha\cdot\mathbf{P}\overline{\mathbf{u}}^\beta\right)\left(\mathbf{v}^\alpha\cdot\overline{\mathbf{v}}^\beta\right)\right]. \qquad (23)$$

# D  TRANSFER LEARNING DERIVATIONS & DETAILS

**Thm 1 (Transfer theorem)** *The transfer benefit $\mathcal{T}^{A\leftarrow B}$:*

- *Is unaffected by the $\overline{\mathbf{U}}^A$ and $\overline{\mathbf{U}}^B$.*

- *Is completely determined by only $\sigma_z^2$, $\overline{\mathbf{S}}^A$, $\overline{\mathbf{S}}^B$, and the $\overline{N_2}^A$ by $\overline{N_2}^B$ similarity matrix $\overline{\mathbf{Q}} = \overline{\mathbf{V}}^{AT}\overline{\mathbf{V}}^B$.*

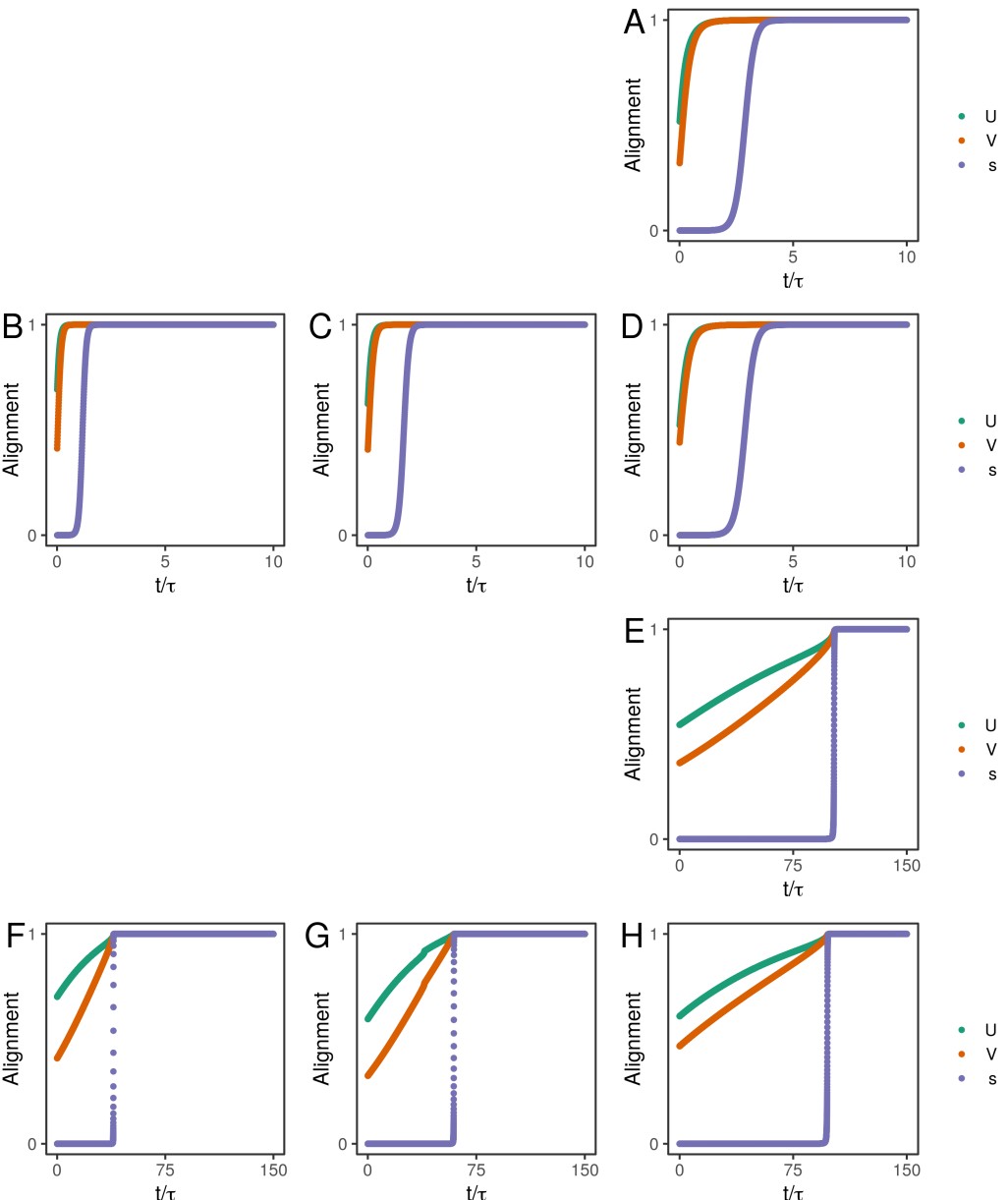

Figure 9: Alignment of randomly-initialized network modes to data modes and growth of singular values, plotted for 1 hidden layer (first two rows, a-d) and 3 hidden layers (last two rows, e-h), and for a rank 1 teacher (first and third rows, a & e), or a rank 3 teacher (second and fourth rows, b-d & f-h). The columns are the different modes, with respective singular values of 6, 4, and 2. $\sigma_z$ was set to 1. The deeper networks show substantially slower mode alignment, with alignment not completed until around when the singular value increases.

**Proof:** We define

$$\overline{\mathbf{U}}^{AB} = \begin{array}{c} \\ \overline{N}_3 \\ \overline{N}_3 \end{array} \begin{array}{cc} \overline{N}_2^A & \overline{N}_2^B \\ \left[ \begin{array}{c|c} \overline{\mathbf{U}}^A & \mathbf{0} \\ \hline \mathbf{0} & \overline{\mathbf{U}}^B \end{array} \right] \end{array} \qquad \overline{\mathbf{S}}^{AB} = \begin{array}{c} \\ \overline{N}_2^A \\ \overline{N}_2^B \end{array} \begin{array}{cc} \overline{N}_2^A & \overline{N}_2^B \\ \left[ \begin{array}{c|c} \overline{\mathbf{S}}^A & \mathbf{0} \\ \hline \mathbf{0} & \overline{\mathbf{S}}^B \end{array} \right] \end{array}$$

$$\overline{\mathbf{V}}^{AB} = \begin{array}{c} \\ \overline{N}_1 \end{array} \begin{array}{cc} \overline{N}_2^A & \overline{N}_2^B \\ \left[ \overline{\mathbf{V}}^A \mid \overline{\mathbf{V}}^B \right] \end{array}$$

$$\overline{\mathbf{W}}^{A+B} = \left[ \begin{array}{c|c} \overline{\mathbf{U}}^A & \mathbf{0} \\ \hline \mathbf{0} & \overline{\mathbf{U}}^B \end{array} \right] \left[ \begin{array}{c|c} \overline{\mathbf{S}}^A & \mathbf{0} \\ \hline \mathbf{0} & \overline{\mathbf{S}}^B \end{array} \right] \left[ \begin{array}{c} \overline{\mathbf{V}}^{A^T} \\ \hline \overline{\mathbf{V}}^{B^T} \end{array} \right] \tag{24}$$

Because of the 0 blocks in $\overline{\mathbf{U}}^{AB}$, the vectors in blocks corresponding to task $A$ and task $B$ are completely orthogonal, so $\overline{\mathbf{U}}^{AB}$ remains orthonormal. Thus the relationship between the $\overline{\mathbf{U}}^A$ and $\overline{\mathbf{U}}^A$ is **irrelevant** to the transfer. (In our simulations we use arbitrary orthonormal matrices for $\overline{\mathbf{U}}^A$ and $\overline{\mathbf{U}}^B$.) Therefore the transfer effects will be entirely driven by the relationship between the matrices $\overline{\mathbf{V}}^A$ and $\overline{\mathbf{V}}^B$ and the singular values.

We define $\overline{N}_2^A$ by $\overline{N}_2^B$ similarity matrix $\overline{\mathbf{Q}} = \overline{\mathbf{V}}^{A^T}\overline{\mathbf{V}}^B$. If we think of the columns of each $\overline{\mathbf{V}}$ as spanning a low dimensional feature space in $\overline{N}_1$ dimensional input space that is important for each task, then $\overline{\mathbf{Q}}$ reflects the input feature subspace similarity matrix. We can now calculate the singular values of $\overline{\mathbf{W}}^{A+B}$. First, note that the input singular modes of $\overline{\mathbf{W}}^{A+B}$ are eigenvectors of $\overline{\mathbf{W}}^{A+B^T}\overline{\mathbf{W}}^{A+B}$, and the associated singular values are square roots of the eigenvalues of $\overline{\mathbf{W}}^{A+B}$. Now

$$\overline{\mathbf{W}}^{A+B^T}\overline{\mathbf{W}}^{A+B} = \overline{\mathbf{V}}^{AB}\overline{\mathbf{S}}^{AB}\overline{\mathbf{U}}^{AB^T}\overline{\mathbf{U}}^{AB}\overline{\mathbf{S}}^{AB}\overline{\mathbf{V}}^{AB^T} = \overline{\mathbf{V}}^{AB}\overline{\mathbf{S}}^{AB2}\overline{\mathbf{V}}^{AB^T}$$

Now if $\vec{c}$ is an eigenvector of this matrix:

$$\overline{\mathbf{V}}^{AB}\overline{\mathbf{S}}^{AB2}\overline{\mathbf{V}}^{AB^T}\vec{c} = \lambda\vec{c}$$

This implies that

$$\overline{\mathbf{V}}^{AB^T}\overline{\mathbf{V}}^{AB}\overline{\mathbf{S}}^{AB2}\overline{\mathbf{V}}^{AB^T}\vec{c} = \lambda\overline{\mathbf{V}}^{AB^T}\vec{c}$$

Hence eigenvalues of $\overline{\mathbf{V}}^{AB}\overline{\mathbf{S}}^{AB2}\overline{\mathbf{V}}^{AB^T}$ are also eigenvalues of $\overline{\mathbf{V}}^{AB^T}\overline{\mathbf{V}}^{AB}\overline{\mathbf{S}}^{AB2}$, with the mapping between the eigenvectors given by $\overline{\mathbf{V}}^{AB}$. Furthermore, this mapping must be a bijection for eigenvectors with non-zero eigenvalues, since the matrices have the same rank (the rank of $\overline{\mathbf{V}}^{AB}$). To see this, note that $\overline{\mathbf{S}}^{AB2}$ is full rank. From this, it is clear that

$$\mathrm{rank}\overline{\mathbf{V}}^{AB^T}\overline{\mathbf{V}}^{AB}\overline{\mathbf{S}}^{AB2} = \mathrm{rank}\overline{\mathbf{V}}^{AB^T}\overline{\mathbf{V}}^{AB} = \mathrm{rank}\overline{\mathbf{V}}^{AB}.$$

Furthermore, $\overline{\mathbf{S}}^{AB2}$ is positive definite, so

$$\mathrm{rank}\overline{\mathbf{V}}^{AB}\overline{\mathbf{S}}^{AB2}\overline{\mathbf{V}}^{AB^T} = \mathrm{rank}\overline{\mathbf{V}}^{AB}.$$

Now that we know the eigenvectors of these matrices are in bijection, note that:

$$\overline{\mathbf{V}}^{AB^T}\overline{\mathbf{V}}^{AB}\overline{\mathbf{S}}^{AB2} = \left[ \begin{array}{c} \overline{\mathbf{V}}^{A^T} \\ \hline \overline{\mathbf{V}}^{B^T} \end{array} \right] \left[ \overline{\mathbf{V}}^A \mid \overline{\mathbf{V}}^B \right] \left[ \begin{array}{c|c} \overline{\mathbf{S}}^{A2} & \mathbf{0} \\ \hline \mathbf{0} & \overline{\mathbf{S}}^{B2} \end{array} \right] = \left[ \begin{array}{cc} \mathbf{I} & \mathbf{Q} \\ \mathbf{Q}^{\mathbf{T}} & \mathbf{I} \end{array} \right] \left[ \begin{array}{c|c} \overline{\mathbf{S}}^{A2} & \mathbf{0} \\ \hline \mathbf{0} & \overline{\mathbf{S}}^{B2} \end{array} \right]$$

Because the output modes don't matter (as noted above), the alignment between the eigenvectors of $\overline{\mathbf{V}}^{AB}\overline{\mathbf{S}}^{AB2}\overline{\mathbf{V}}^{AB^T}$ and $\overline{\mathbf{V}}^A$, weighted by their respective eigenvalues, gives the transfer benefit.

For any given tasks, the transfer benefit can be calculated using our theory. However, in certain special cases, we can give exact answers. For example, in the rank one case with equal singular values between the tasks ($\overline{s}_A = \overline{s}_B = \overline{s}$), the matrix

$$\begin{bmatrix} \mathbf{I} & \mathbf{Q} \\ \mathbf{Q^T} & \mathbf{I} \end{bmatrix} \left[ \begin{array}{c|c} \overline{\mathbf{S}}^{A2} & \mathbf{0} \\ \hline \mathbf{0} & \overline{\mathbf{S}}^{B2} \end{array} \right]$$

reduces to

$$\begin{bmatrix} 1 & q \\ q & 1 \end{bmatrix} \overline{s}^2$$

with eigenvalues $s\sqrt{1 \pm q}$ and eigenvectors

$$\begin{bmatrix} 1 & 1 \\ 1 & -1 \end{bmatrix}$$

Corresponding to the shared structure between the tasks and the differences between them. We note that the sign of the alignment $q$ is irrelevant as a special case of the fact (noted above) that any orthogonal transformation on the output modes does not affect transfer.

### D.1 Misalignment and interference

Why is there interference between tasks which are not well aligned? In the rank one case, we are effectively changing the (input) singular dimensions of $\overline{\mathbf{Y}}_A$ from $\overline{\mathbf{V}}^A$ to $\overline{\mathbf{V}}^{AB}$. The two singular modes of $\overline{\mathbf{V}}^{AB}$ correspond to the shared structure between the tasks (weighted by the relative signal strengths), and the differences between them, respectively. Although we may be improving our estimates of the shared mode if $q > 0$ (by increasing its singular value relative to $\overline{s}_A$), we are actually decreasing its alignment with $\overline{\mathbf{V}}^A$ unless $q = 1$. This misalignment is captured by the second mode of $\overline{\mathbf{V}}^{AB}$, but the *increase* in the singular value of the first mode must come at the cost of a *decrease* in the singular value of the second mode. See Fig. 10 for a conceptual illustration of this. This means that the multi-task setting allows the distinctions between the tasks to sink towards the sea of noise, while pulling out the common structure. In other words, transferring knowledge from one task always comes at the cost of ignoring differences between the tasks. Furthermore, incorporating a task $B$ allows its noise to seep into the task $A$ signal. Together, these two effects help to explain why transfer can be sometimes beneficial but sometimes detrimental.

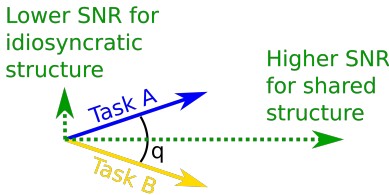

Figure 10: Conceptual cartoon of how $\mathcal{T}^{A \leftarrow B}$, the transfer benefit (or cost) arises from alignment between the task's input modes.

## E  Non-gradient training algorithm

In Fig. 11 we show the match between the error achieved by training the student by gradient descent and the optimal stopping error predicted by the non-gradient shrinkage algorithm in the case of a rank-1 teacher.

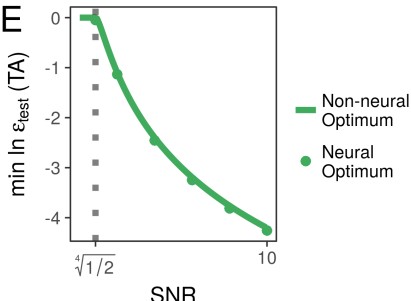

Figure 11: Match between optimal stopping error prediction from non-gradient training algorithm and empirical optimal stopping error for a rank-1 teacher.

## F    TRANSFER RESULTS GENERALIZE TO NON-LINEAR NETWORKS

Since most deep learning practitioners do not train linear networks, it is important that our theoretical insights generalize beyond this simple case. In this section we show that the transfer patterns qualitatively generalize to non-linear networks.

Here, we show results from teacher networks with $\overline{N}_1 = 100$ $\overline{N}_3 = 50$, $\overline{N}_2 = 4$ (thus the task is higher rank) and leaky relu non-linearities at the hidden and output layers. We train a student with leaky relu units and $N_2 = N_3$ to solve this task. Results qualitatively look quite similar to those in Fig 5. of the main text for rank one linear teachers, see below. Thus our insights into transfer may help to understand multi-task benefits in more complicated architectures.

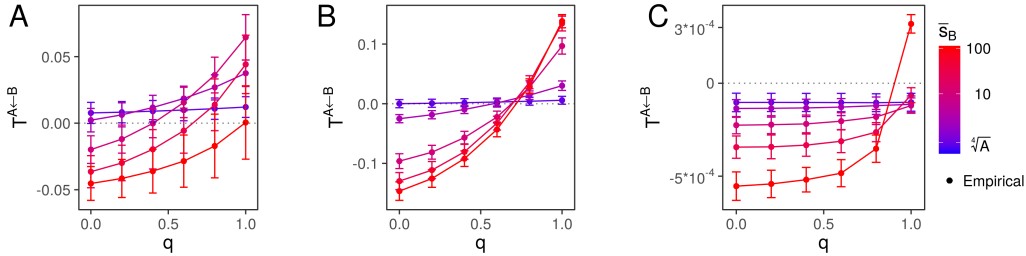

Figure 12: Transfer benefit $\mathcal{T}^{A \leftarrow B}(\overline{s}_A, \overline{s}_B, q)$ for non-linear teachers and students, plotted at different values of $\overline{s}_A$. (a) $\overline{s}_A = 0.84 = \sqrt[4]{\mathcal{A}}$. With support from another aligned task, especially one with moderately higher SNR, performance on a low SNR task will improve. (b) $\overline{s}_A = 3$. Tasks with modest signals will face interference from poorly aligned tasks, but benefits from well aligned tasks. These effects are amplified by SNR. (c) $\overline{s}_A = 100$. Tasks with very strong signals will show little effect from other tasks (note y-axis scale), but any impact will be negative unless the tasks are very well aligned.

## G    VARYING THE NUMBER OF TRAINING EXAMPLES

In the main text, we focused on the test error dynamics in the case in which the number of examples equalled the number of inputs. Here we show how the formula for test error curves is modified as the number of training examples $P$ is varied. For simplicity, when $P \neq N_1$, we focus on the case of a full rank student with aspect ratio $\mathcal{A} = 1$ (so that $N_1 = N_2 = N_3$). The more general case of lower rank students with non-unity aspect ratios can be easily found from this case, but with some additional bookkeeping.

As before, we assume the teacher generates noisy outputs from a set of $P$ inputs:

$$\hat{\mathbf{y}}^\mu = \overline{\mathbf{W}}\hat{\mathbf{x}}^\mu + \mathbf{z}^\mu \qquad \text{for} \quad \mu = \mathbf{1}, \ldots, \mathbf{P}. \tag{25}$$

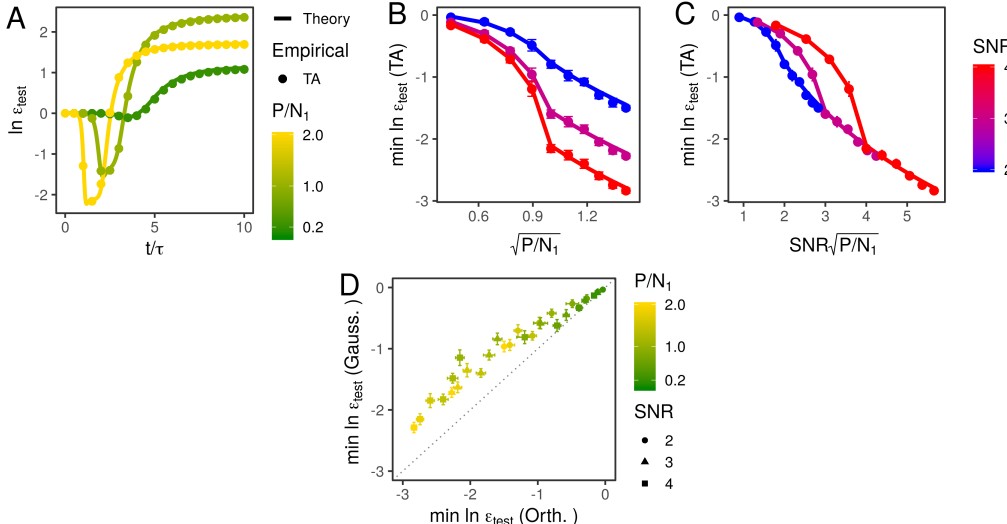

Figure 13: The effects of varying the number of training examples $P$. (a) Test error for a student learning from a rank-1 teacher with an SNR of 3, with different numbers of inputs. (b,c) Minimum generalization error plotted against $\sqrt{P/N_1}$ and SNR $\cdot \sqrt{P/N_1}$, respectively, at different SNRs. When $P \geq N_1$, the minimum generalization error is simply determined by SNR$\sqrt{P/N_1}$, so all curves converge to a single asymptotic line in (c) as $P$ increases. When $P < N_1$, however, the curves for different SNRs separate because the projection and noise effects depend on initial SNR. (d) Optimal stopping error for gaussian vs. orthogonal inputs, showing a strong correlation. Thus our use of orthogonal inputs in the theory also yields insight into the more general case of approximately unit norm Gaussian inputs. (For all panels $N_1 = N_2 = N_3 = 100, \overline{N}_2 = 1$.)

This training set yields important second-order training statistics that will guide student learning:

$$\mathbf{\Sigma}^{11} \equiv \hat{\mathbf{X}}\hat{\mathbf{X}}^T \qquad \mathbf{\Sigma}^{31} \equiv \hat{\mathbf{Y}}\hat{\mathbf{X}}^T = \overline{\mathbf{W}}\hat{\mathbf{X}}\hat{\mathbf{X}}^T + \mathbf{Z}\hat{\mathbf{X}}^T. \tag{26}$$

Here $\hat{\mathbf{X}}$, $\hat{\mathbf{Y}}$, and $\mathbf{Z}$ are each $\overline{N}_1$ by $P$, $\overline{N}_3$ by $P$, and $\overline{N}_3$ by $P$ matrices respectively, whose $\mu$'th columns are $\hat{\mathbf{x}}^\mu$, $\hat{\mathbf{y}}^\mu$, and $\hat{\mathbf{z}}^\mu$, respectively. $\mathbf{\Sigma}^{11}$ is an $\overline{N}_1$ by $\overline{N}_1$ input correlation matrix, and $\mathbf{\Sigma}^{31}$ is an $\overline{N}_3$ by $\overline{N}_1$ the input-output correlation matrix. We choose the matrix elements $z_i^\mu$ of the noise matrix $\mathbf{Z}$ to be drawn iid from a Gaussian with zero mean and variance $\sigma_z^2/\overline{N}_1$. The noise scaling is chosen so the singular values of the teacher $\overline{\mathbf{W}}$ and the noise $\mathbf{Z}$ are both $O(1)$, leading to non-trivial generalization effects. Furthermore, we chose training inputs to be close to unit-norm, and make the input covariance matrix $\mathbf{\Sigma}^{11}$ as white as possible (whitening is a common pre-processing step for inputs). When $P > \overline{N}_1$, this can be done by choosing the *rows* of $\hat{\mathbf{X}}$ to be orthonormal and then scaling up by $\sqrt{P/\overline{N}_1}$, so the columns are approximately unit norm. Then $\mathbf{\Sigma}^{11} = P/\overline{N}_1\mathbf{I}$ is proportional to the identity. On the otherhand, if $P < \overline{N}_1$, we choose the columns of $\hat{\mathbf{X}}$ to be orthonormal, so that $\mathbf{\Sigma}^{11} = \mathcal{P}^\|$, where $\mathcal{P}^\|$ is a projection operator onto the $P$ dimensional column space of $\hat{\mathbf{X}}$ spanned by the input examples. Both these choices are intended to approximate the situation in which the columns of $\hat{\mathbf{X}}$ are chosen to be iid unit-norm vectors. Finally, as generalization performance will depend on the *ratio* of teacher singular values to the noise variance parameter $\sigma_z^2$, we simply set $\sigma_z = 1$ as in the main text. Thus, given the unit-norm inputs, we can think of the teacher singular values as signal to noise ratios (SNRs). We now examine how the dynamics of the test error evolves as we vary the number of training examples $P$. We split our analyses into two distinct regimes: (1) the oversampled regime in which the data density $\mathcal{D} \equiv P/N_1 > 1$, and (2) the undersampled regime in which $\mathcal{D} < 1$.

### G.1 THE OVERSAMPLED REGIME

The oversampled regime ($\mathcal{D} > 1$) is relatively simple. First $\mathbf{\Sigma}^{11}$ is scaled up by a factor of $\mathcal{D}$. And in the input-output covariance matrix, $\mathbf{\Sigma}^{31} = \overline{\mathbf{W}}\hat{\mathbf{X}}\hat{\mathbf{X}}^T + \mathbf{Z}\hat{\mathbf{X}}^T$, the signal component, $\overline{\mathbf{W}}\hat{\mathbf{X}}\hat{\mathbf{X}}^T$ is

scaled up by a factor of $\mathcal{D}$ while the noise component $\mathbf{Z}\hat{\mathbf{X}}^T$ has the same singular value spectrum as the $\mathcal{D} = 1$ case, up to an overall scaling by $\sqrt{\mathcal{D}}$ (since the rows of $\hat{\mathbf{X}}$ are orthogonal and all its singular values are equal to $\sqrt{\mathcal{D}}$). This leads to an increase in the effective SNR by a factor of $\sqrt{\mathcal{D}}$. Thus overall, the test error curves for the case of $\mathcal{D} > 1$ can be simply obtained from the theory of the test error curves for $\mathcal{D} = 1$ through two modifications: (1) a boost in the SNR for the $\mathcal{D} = 1$ case by a multiplicative factor of $\sqrt{\mathcal{D}}$, and (2) and an overall speed up in the learning time by a multiplicative factor of $\mathcal{D}$.

### G.2 THE UNDERSAMPLED REGIME

For the undersampled regime ($\mathcal{D} < 1$), we must account for the fact that the $P$ training inputs do not span the full $N_1$ dimensional space of all inputs. Thus the projection operator $\mathcal{P}^{\|}$ onto the $P$ dimensional column space of $\hat{\mathbf{X}}$ plays a crucial role. Indeed the input-correlation $\mathbf{\Sigma}^{11} = \mathcal{P}^{\|}$. And $\mathbf{\Sigma}^{31} = \overline{\mathbf{W}}\mathcal{P}^{\|} + \mathbf{Z}\hat{\mathbf{X}}^T$. This implies that the learning dynamics only transforms the composite student map $\mathbf{W}$ from the $P$ dimensional subspace spanned by the inputs to the $N_3$ dimensional output space. In contrast, the student map from the $N_1 - P$ dimensional subspace orthogonal to the image of $\mathcal{P}^{\|}$ remains frozen. Tracing through the equations of the main paper and accounting for the projection operator $\mathcal{P}^{\|}$, we find the effective aspect ratio for this undersampled learning problem (when $N_3 = N_2 = N_1$) is no longer $\mathcal{A} = N_3/N_1$ but rather $\mathcal{D} = P/N_1$. Furthermore, in the limit $\overline{N}_3, \overline{N}_1 \to \infty$ while $\overline{N}_2$ remains $O(1)$, the singular values of the signal component $\overline{\mathbf{W}}\mathcal{P}^{\|}$ of $\mathbf{\Sigma}^{31}$ are attenuated by a factor of $\sqrt{\mathcal{D}}$, making the associated singular vectors more susceptible to noise. Again tracing through the equations of the main paper, with all of these modifications, we find the final formula for test error curves in the undersampled measurement regime:

$$\varepsilon_{\text{test}}(t) = \frac{\left[(N_3 - P)\epsilon^2 + (P - \overline{N}_2)\langle s(\hat{s}, t)^2\rangle + \sum_{\alpha=1}^{\overline{N}_2}\left[(s_\alpha(t) - \overline{s}_\alpha)^2 + 2s_\alpha(t)\overline{s}_\alpha(1 - \mathcal{O}(\sqrt{\mathcal{D}}\overline{s}_\alpha))\right]\right]}{\left[\sum_{\alpha=1}^{\overline{N}_2}\overline{s}_\alpha^2\right]} \tag{27}$$

This equation has several modifications compared to the case $P = N_1$ in (15). First the term in the numerator involving $N_3 - P$ reflects generalization error due to the $N_3 - P$ dimensional frozen subspace, and the initial weight variance $\epsilon^2$ contributes to this generalization error. The second term in the numerator involves all the $P - \overline{N}_2$ training modes which cannot be correlated with the teacher, and the average $\langle\cdot\rangle$ is over a Marcenko-Pasteur distribution of singular values (see (13)) except with the aspect ratio $\mathcal{A}$ replaced by $\mathcal{D}$. The third term accounts for learned correlations between the student and teacher. It involves the transformation from teacher singular values $\overline{s}$ to training data singular values $\hat{s}$ through the formula (11) except with the aspect ratio replacement $\mathcal{A} \to \mathcal{D}$, and the effective teacher singular value attenuation $\overline{s} \to \sqrt{\mathcal{D}}\overline{s}$. Similarly, the computation of the singular vector overlap is done through (12) also with the replacements $\mathcal{A} \to \mathcal{D}$ and $\overline{s} \to \sqrt{\mathcal{D}}\overline{s}$.

### G.3 COMPARISON OF THEORY AND EXPERIMENT FOR UNDER AND OVER SAMPLED MEASUREMENT REGIMES

In Fig. 13, we show an excellent match between our theory and empirical simulations for varying values of $P$, both in the oversampled and undersampled measurement regimes. There are a number of interesting features to note. First, although the minimum generalization error improves monotonically with $P$, the asymptotic ($t \to \infty$) generalization error does not, because of a *frozen subspace* (Advani & Saxe, 2017) of the modes that are not overfit when $P < N_1$, because the training data rank is $\leq P$. Second, when $P \geq N_1$, the minimum generalization error is simply determined by SNR$\sqrt{P/N_1}$, so all curves converge to a single asymptotic line as $P$ increases. When $P < N_1$, however, the curves for different SNRs separate because the projection and noise effects depend on initial SNR. Finally, in Fig. 13D we show that approximately unit norm i.i.d. gaussian inputs yield similar results to the orthogonalized data matrices we employed in the theory, although the gaussian inputs do result in slightly higher optimal stopping error.

## H LESS THAN FULL RANK STUDENTS

Although we generally assumed students were full rank in the main text to simplify the calculations, our theory remains exact for TA networks of any rank. Furthermore, as shown in Fig. 14, the TA

and random networks again show very similar optimal stopping generalization error, but with the optimal stopping time of the random networks lagging behind that of the TA networks. Furthermore, this lag increases as the rank of the random network decreases (because a low rank network will have less initial projection onto the random modes, there is is more alignment to be done). However, reducing the student rank does not change the optimal stopping error (as long as it is still greater than the teacher rank).

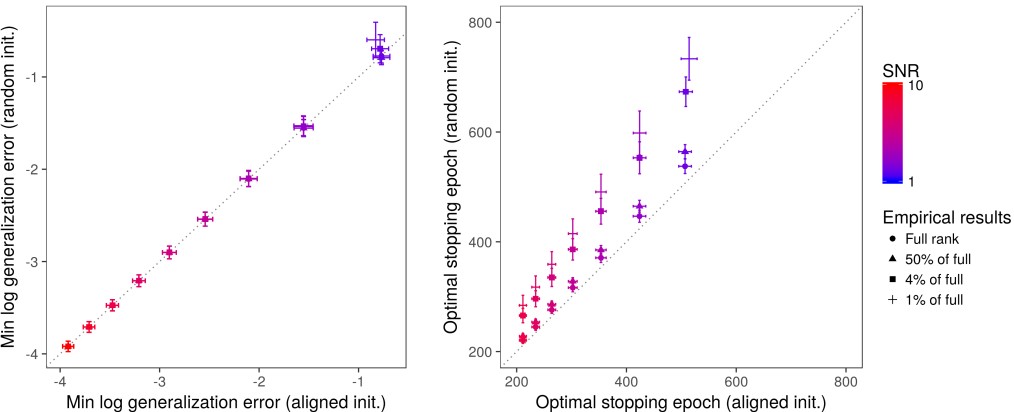

(a) Best generalization error is quite similar between aligned and random initializations

(b) Optimal stopping time is quite similar between aligned and random initializations

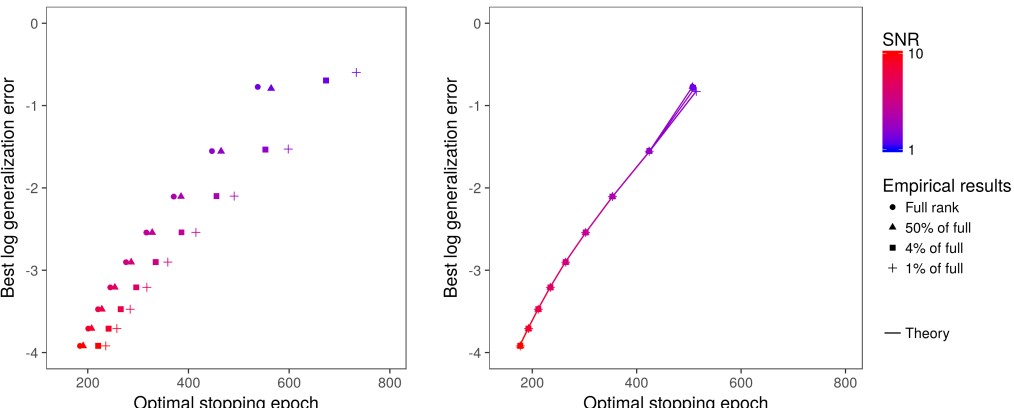

(c) Optimal generalization error vs. optimal stopping time for randomly initialized networks

(d) Optimal generalization error vs. optimal stopping time for initially aligned networks

Figure 14: Empirical verification that the simplifying assumptions of our theory are approximately valid in the regime we are considering at different student ranks. Initializations with random initial weights (random init.) and initializations with initial weight aligned to the noisy data SVD (aligned init.) are compared across varying student ranks. (a) The minimum generalization errors are almost identical between the different initializations and different student ranks. (b) The optimal stopping time in the randomly initialized networks consistently lags behind the aligned networks, because it takes time for the alignment to occur. This lag increases as the students rank decreases. (c) Randomly initialized networks of varying ranks obey qualitatively similar trends of increase in optimal stopping error and optimal stopping time as SNR decreases. (d) The theory predicts the aligned networks trends of increase in optimal stopping error and optimal stopping time with decreasing SNR almost perfectly. (All plots are made with a rank 1 teacher and $N_1 = N_3 = 100$)

