# OpenReview forum: "An analytic theory of generalization dynamics and transfer learning in deep linear networks"
_ICLR.cc/2019/Conference_

### Official Review · AnonReviewer3 · 2018-11-03
**Interesting analysis of deep linear networks and task structure**

**Rating:** 6
**Confidence:** 3

**Review:**

This paper builds on the long recent tradition of analyzing deep linear neural networks. In addition to an ample appendix bringing the page total to 20, the authors went over the recommended eight pages, hitting the hard limit of 10 and thus per reviewing directions will be held to a higher standard than the other (mostly 8-page) papers.

The recent literature on deep linear networks has explored many paths with the hope of producing insights that might help explain the performance of deep neural networks. A recent line of papers by Soudry and Srebro among others focuses on the behavior of stochastic gradient descent. This paper’s analysis comes from a different angle, following the work by Saxe et al (2013) whose analysis considers a (classic one hidden layer) linear teacher network that generates labels and a corresponding student trained to match those labels. The analysis hinges on the singular value decomposition of the composite weight matrix USV^T = W = W^{32} W^{21}.

One aim of the present work, that appears to be a unique contribution above the prior work is to focus on the role played by task structure, suggesting that certain notions of task structure may play a more significant role than architecture and that any bounds which consider architecture but not task structure are doomed to be excessively loose.

To facilitate their analysis, the authors consider an artificial setup that requires some specific properties. For example, the number of inputs are equal to the input dimension of the network, with the inputs themselves being orthonormal. The labeling function includes a noise term and the singular values of the teacher model admit an interpretation as signal to noise ratios. Given their setup, the authors can express the train and test errors analytically in terms of the weight matrices of the teacher and student and the input-output covariance matrix. The authors then analyze the gradient descent dynamics in what appears to follow the work of Saxe 2013 although I am not an expert on that paper. The analysis focuses on the time dependent evolution of the singular values of the student model, characterized via a set of differential equations.

The next analysis explores a condition that the authors dub “training aligned” initial conditions. This involves initializing the student weights to have the same singular vectors as the training data input-output covariance but with all singular values equal to some amount epsilon. The authors show that the learning dynamics give rise to what they characterize as a singular value “detection wave”. Detecting the modes in descending order by their corresponding singular values.

A set of synthetic experiments show close alignment between theory and experiment.

Section 3.5 offers just one paragraph on a “qualitative comparison to nonlinear networks”. A few issues here are that aesthetically, one-paragraph subsections are not ideal. More problematic is that this theory presumably is building towards insights that might actually be useful towards understanding deep non-linear networks. Since the present material is only interesting as an analytic instrument, I would have hoped for greater emphasis on these connections, with perhaps some hypotheses about the behavior of nonlinear nets driven by this analysis that might subsequently be confirmed or refuted.

The paper concludes with two sections discussing what happens when nets are trained on randomly labeled data and knowledge transfer across related tasks respectively.

Overall I think the paper is well-written and interesting, and while I haven’t independently verified every proof, the technical analysis appears to be interesting and sound. The biggest weaknesses of this paper---for this audience, which skews empirical---concern the extent to which the work addresses or provides insight about real neural networks. One potential weakness in this line of work may be that it appears to rely heavily on the linearity of the deep net. While some other recent theories seem more plausibly generalized to more general architectures, it’s not clear to me how this analysis, which hinges so crucially on the entire mapping being expressible as a linear operator, can generalize.

On the other hand, I am personally of the opinion that the field is in the unusual position of possessing too many tools that “work” and too few new ideas. So I’m inclined to give the authors some license, even if I’m unsure of the eventual utility of the work.

One challenge in reviewing this paper is that it builds tightly on a number of recent papers and without being an authority on the other works, while it’s possible to assess the insights in this paper, it’s difficult to say conclusively which among them can rightly be considered the present paper’s contributions (vs those of the prior work).

---

> ### Author Response · Authors · 2018-11-26
> **Thanks, some responses and revisions**
>
> Thank you for the thorough and helpful comments. We appreciate the time and care you took reviewing this paper. We have responded to a few points below.
>
> “One aim of the present work, that appears to be a unique contribution above the prior work is to focus on the role played by task structure, suggesting that certain notions of task structure may play a more significant role than architecture and that any bounds which consider architecture but not task structure are doomed to be excessively loose.” — Thank you for highlighting this, because we think it is important for thinking about developing better generalization bounds for deep learning systems.
>
> Nonlinear networks: Our hope is that most the understanding here will actually (qualitatively) generalize to the nonlinear case. We hope that therefore the present results will not  only be useful analytically, but will also provide insights into the puzzles in the nonlinear case that may yield new directions for research in nonlinear generalization and regularization. We maybe should have communicated this more clearly, and have made some minor revisions in order to do so. In particular:
>
> a) We have combined the qualitative comparison with nonlinear networks with the rest of the theory verification section, both because of the aesthetic issue you noted, and to avoid the appearance that this was the only portion of the paper that is relevant to the nonlinear case.
>
> b) The issues about memorization capacity vs. observed generalization were observed in nonlinear networks originally. We explain this in the linear case in terms of task structure in the data being stronger than noise structure, and therefore being learned first. We suggest that the phenomenon has similar causes in nonlinear networks, and this motivates the argument that the many data-agnostic bounds will be very loose in the nonlinear case (as we have shown they are in the linear case, and as they have been observed to be in the nonlinear case).
>
> c) The related issue about memorizing randomized labels taking longer than learning the real data also are similar in nonlinear networks, in fact that was our inspiration for exploring this.
>
> d) Our results on transfer qualitatively generalize to the nonlinear case, as shown in the supplementary material (Appendix F). In particular, we observe the same pattern of interference between weakly aligned tasks and benefits between well-aligned ones, modulated in similar ways by the relative SNRs of the tasks. We reworded part of the transfer section in an attempt to highlight this further.
>
> e) While our non-gradient optimal learning algorithm will clearly not work in the nonlinear case, we think it suggests a useful direction for future research: more structure-sensitive regularization strategies may substantially outperform relatively naive ones like Lp penalties.
>
> f) We have edited the discussion section to highlight some of the above points.
>
> Finally, regarding the comment that "the number of inputs are equal to the input dimension of the network, with the inputs themselves being orthonormal." In the original paper we actually showed how to handle situations where the number of inputs is both less than and greater than the input dimension of the network in Appendix G and we verified the match between our theory and experimental results in Figure 13.  We mentioned this briefly in the main paper in Sec. 2.1, and we apologize if it wasn't clear.
>
> Thanks again for your comments!

---

### Official Review · AnonReviewer1 · 2018-11-05
**study of deep linear networks towards understanding generalization and transfer learning**

**Rating:** 7
**Confidence:** 4

**Review:**

Most theoretical work on understanding generalization in deep learning provides very loose bounds and does not adequately explain this phenomena. Moreover, their is also a gap in our understanding of knowledge transfer across tasks.

The authors study a simple model of linear networks. They give analytic bounds on train/test error of linear networks as a function of training time, number of examples, network size, initialization, task structure and SNR. They argue that their results indicate that deep networks progressively learn the most important task structure first, so the generalization at the early stopping primarily depends on task structure and is independent of network size. This explains previous observations about real data being learned faster than random data. Interestingly, they show a learning algorithm that provably out-performs gradient-descent training. They show that how knowledge transfer in their model depends on SNRs and input feature alignments of task-pairs.

The theoretical framework of low-rank noisy teachers using a more complex (e.g., wider or deeper) student network is simple and allows them to use random matrix theory to understand and interpret  interesting scenarios such as random initialization vs. training-aligned initialization. Fig. 5 shows that even though the theoretical framework is for linear networks, many of the observations hold even for non-linear (leaky ReLU) networks. They also offer a reasonable explanation for learning random vs. real data in terms of how the signal singular values get diluted or spread over many modes.

I do not think that the generalization bounds given by the authors are any tighter than previous attempts. However, I think their theoretical and experimental contributions to provide quantitative and qualitative explainations of various interesting phenomena in deep learning are both solid enough to merit acceptance.

---

> ### Author Response · Authors · 2018-11-26
> **Thanks for your comments, some clarification about the bounds**
>
> Thank you for the positive comments. We think this provides a useful synopsis of the contributions of our paper.
>
> With respect to the tightness of generalization bounds, nobody to the best of our knowledge has given exact analytic estimates for optimal stopping error in the linear case with rank > 1 (asymptotic is easier, since it is simply the least-squares solution). Thus this is a new (and tighter) bound in the limited setting we consider. However, we primarily see this work as providing more general insights into why naive generalization bounds are loose — if the structure of the data is a primary factor driving generalization performance in the nonlinear case (as we have shown it is for linear networks), then bounds based solely on architecture are likely to be useless (as they have generally been observed to be for deep learning models). This is also related to the memorization time issue we discuss.

---

### Official Review · AnonReviewer2 · 2018-11-06
**interesting and thorough theory of generalization and transfer learning**

**Rating:** 8
**Confidence:** 4

**Review:**

Pros: The paper tackles an interesting problem of generalization and transfer learning in deep networks. They start from a linear network to derive the theory, identifying phases in the learning, and relating learning rates to task structure and SNR. The theory is thorough and backed up by numerical simulations, including qualitative comparisons to nonlinear networks.

The intuition behind alignment of tasks

Cons: Most of the theory is developed on a linear network in an abstracted teacher/student/TA framework, where the analysis revolves around the the SVD of the weights. It's unclear to what extent the theory would generalize not only to deep, nonlinear networks (which the paper addresses empirically) but also different structures in the task that are not well approximated by the SVD.

---

> ### Author Response · Authors · 2018-11-26
> **Thanks, some responses and revisions**
>
> Thank you for the helpful and positive comments.
>
> Nonlinear networks: Our hope is that most the understanding here will actually (qualitatively) generalize to the case of nonlinear networks learning nonlinear structure (i.e. structure which is not captured by the SVD). We hope that therefore the present results will not  only be useful analytically, but will also provide insights into the puzzles in the nonlinear case that may yield new directions for research in nonlinear generalization and regularization. We maybe should have communicated this more clearly, and have made some minor revisions in order to do so, since both you and reviewer 3 seemed to want more clarification of this issue. In particular:
>
> a) We have combined the qualitative comparison with nonlinear networks with the rest of the theory verification section, both because of the aesthetic issue you noted, and to avoid the appearance that this was the only portion of the paper that is relevant to the nonlinear case.
>
> b) The issues about memorization capacity vs. observed generalization were observed in nonlinear networks originally. We explain this in the linear case in terms of task structure in the data being stronger than noise structure, and therefore being learned first. We suggest that the phenomenon has similar causes in nonlinear networks, and this motivates the argument that the many data-agnostic bounds will be very loose in the nonlinear case (as we have shown they are in the linear case, and as they have been observed to be in the nonlinear case).
>
> c) The related issue about memorizing randomized labels taking longer than learning the real data also are similar in nonlinear networks, in fact that was our inspiration for exploring this.
>
> d) Our results on transfer qualitatively generalize to the nonlinear case, as shown in the supplementary material (Appendix F). In particular, we observe the same pattern of interference between weakly aligned tasks and benefits between well-aligned ones, modulated in similar ways by the relative SNRs of the tasks. We reworded part of the transfer section in an attempt to highlight this further.
>
> e) While our non-gradient optimal learning algorithm will clearly not work in the nonlinear case, we think it suggests a useful direction for future research: more structure-sensitive regularization strategies may substantially outperform relatively naive ones like Lp penalties.
>
> f) We have edited the discussion section to highlight some of the above points.

---

### Public Comment · ~Mohammad_Pezeshki1 · 2018-09-28
**A few remarks**

Dear authors,
Interesting work! I very much like the intuition that your paper provides on how the "singular value detection wave" could penetrate into the MP distribution and result in overfitting!

In section 3.2, you mention that the training data singular value (\hat{s}) is greater than the original teacher singular value (\bar{s}). Where you refer to as being inflated by noise. So, why is that the case all the time? One would expect the noise to reduce the singular value, I guess. I appreciate it if you could elaborate on that.

Another thing about the non-gradient-decent optimization method. It would only work in a fully linear network if I understand correctly. Right?

I know that getting an analytical solution for the general case of random networks (not TA) is difficult, but I would like to know your thoughts on how would you expect the singular vectors evolve with random initialization?

Finally, in Eq. 8, I think the second covariance matrix must be \Sigma^{11} and not \Sigma^{31}.

Thanks in advance for your time and I hope this paper get accepted!

---

> ### Author Response · Authors · 2018-09-29
> **Response to your remarks**
>
> Hi Mohammad,
>
> Thanks for your interest! I've responded point by point below:
>
> * In section 3.2, you mention that the training data singular value (\hat{s}) is greater than the original teacher singular value (\bar{s}). Where you refer to as being inflated by noise. So, why is that the case all the time? One would expect the noise to reduce the singular value, I guess. I appreciate it if you could elaborate on that.
>
> The derivation of how the singular vectors and values are distorted is actually quite complex, I encourage you to check out our reference for it (you can find it here https://arxiv.org/pdf/1103.2221.pdf). However, to give you some intuition, imagine a simpler case, where we are keeping the input mode fixed, and just adding i.i.d. noise to the output mode. In this case, in the high-dimensional limit in which we are working, the noise is with very high probability orthogonal to the output mode. Adding two orthogonal vectors results in a longer vector, and this is the basic intuition underlying why the singular values get inflated. But please check out the reference for more details, even if you don't want to follow the full derivation you can get a better intuition by looking at the equations they give on page 2 that approximately give the resulting singular values, and consider increasing the noise from zero.
>
> ** Another thing about the non-gradient-decent optimization method. It would only work in a fully linear network if I understand correctly. Right?
>
> Yes, and it also requires knowledge of either something about the noise structure OR the singular values of the noise-free teacher, in order to figure out the correct shrinking of the singular values. However, we are very interested in the possibility that some generalization of this idea would be applicable more generally.
>
> ** I know that getting an analytical solution for the general case of random networks (not TA) is difficult, but I would like to know your thoughts on how would you expect the singular vectors evolve with random initialization?
>
> We show in the main text simulations for how the train and test error evolve with random initialization. In the supplementary material, appendix B/Fig 9 we show (empirically) exactly what you're asking for: how the singular vectors of a randomly initialized network align over time, in both the shallow and deep case, and how that timing matches up with the timing of the increase in the singular value. In the shallow case, the alignment happens quite fast, long before the singular value increases, but in the deep case the alignment is only completed about when the singular value rises sharply.
>
> ** Finally, in Eq. 8, I think the second covariance matrix must be \Sigma^{11} and not \Sigma^{31}.
>
> Yes, you are absolutely right! Thank you!
>
> ** Thanks in advance for your time and I hope this paper get accepted!
>
> Thanks!

---

### Public Comment · (anonymous) · 2018-10-09
**comments & questions**

This paper presents some useful intuitions about certain generalization phenomena in neural networks, such as the observation that randomly labeled data take more time to learn than correctly labeled data etc. If I may digress a bit here, I also found the language used to express some of the results pleasantly suggestive ("sinking toward the sea of noise"): kudos!

I had a few questions and comments about the paper. I hope they will be useful to the authors. Please feel free to address or ignore them as you see fit.

- You don't treat regularization in this paper, which is an important dimension of generalization in more realistic scenarios. I assume simple l2-norm regularization on the student would manifest itself as some kind of force pulling the student singular values toward zero. How would this affect the analysis? This shouldn't be too difficult to incorporate into the current analysis, I imagine. Other types of regularization (l1-norm or dropout etc) may be more difficult to analyze, but may still be tractable.

- Relatedly, is it possible that just plain old l2-norm regularization on the student would be able to approximate (or even match, with proper tuning) your non-gradient descent based optimal solution to the test error, given that that solution is always smaller than the gradient descent solution (without regularization)?

- I know that you present some results with deeper student networks (Fig. 4). However, the effect of depth is not discussed much. Are there scenarios in your analysis under which increased depth would be beneficial for training speed (as in this paper, for example: https://arxiv.org/abs/1802.06509) or final test error? I'm guessing not.

- It seems to me that in your analysis having a more over-parametrized student (larger N2) can never be beneficial for final test error. This seems to contradict the results in this paper on the effects of over-parametrization on test error (over-parametrization improves test error): https://arxiv.org/abs/1805.12076

 - In practice, early stopping does not really seem to be necessary to prevent overfitting in most cases (for example, see the error curves in this paper: https://arxiv.org/abs/1512.03385). Again, this seems to contradict a main result of this paper. Could you comment on why early stopping does not seem to be needed in practice (regularization?, learning rate adaptation?)?

- For the non-linear results, I'm curious why you chose the leaky-relu, as this seems to be non-standard.  Is it because leaky-relu is "more linear" in some sense? What is the leakage parameter used there? How do the results differ if you use relu instead?

- Finally, I know that this paper considers a rather simple setup, but how do you envision explaining the effects of architecture with the kind of analysis presented in this paper? Architecture choice is probably one of the most important factors determining generalization performance. For image data, for example, going from a fully-connected to a convolutional model probably explains the bulk of the generalization improvement. Additional benefits also come from other architectural improvements like residual connections etc.

---

> ### Author Response · Authors · 2018-10-18
> **Some responses and answers**
>
> Sorry for the slow response, it's been a bit busy here. Thank you for all your comments, including the one about aquatic analogies. :) We appreciate them and endeavor to give a response to all your comments below.
>
> - "You don't treat regularization in this paper, which is an important dimension of generalization in more realistic scenarios. I assume simple l2-norm regularization on the student would manifest itself as some kind of force pulling the student singular values toward zero. How would this affect the analysis? This shouldn't be too difficult to incorporate into the current analysis, I imagine. Other types of regularization (l1-norm or dropout etc) may be more difficult to analyze, but may still be tractable."
>
> Yes, regularization is a very important issue, which we didn't have room to discuss in this paper. We should be able to extend the theory to L2 regularization at least, and we hope to do so in future work if not in a revision to this work.  But your intuition is exactly correct: L2 regularization will shrink the composite teacher singular values towards 0.
>
> - "Relatedly, is it possible that just plain old l2-norm regularization on the student would be able to approximate (or even match, with proper tuning) your non-gradient descent based optimal solution to the test error, given that that solution is always smaller than the gradient descent solution (without regularization)?"
>
> In general, L2 regularization can help.  However, just like optimal early stopping, in the case of teacher ranks > 1 it cannot achieve as good of performance as our optimal solution, because the regularization weight would need to be tuned differently for each teacher mode. This is clear because the optimal shrinkage factor, which can be deduced by inverting Equation 11, depends sensitively on the actual singular value of corresponding mode of the teacher.   One can check that no value of the regularization weight assigned to the L2 penalty can achieve performance equal to our optimal algorithm, just as no early stopping time can achieve this optimal performance either.
>
> - "I know that you present some results with deeper student networks (Fig. 4). However, the effect of depth is not discussed much. Are there scenarios in your analysis under which increased depth would be beneficial for training speed (as in this paper, for example: https://arxiv.org/abs/1802.06509) or final test error? I'm guessing not. "
>
> In fact, the paper you reference noted a *slowdown* with depth in the case of L2 loss, and only saw a speedup in the case of an Lp loss with p > 2. Their L2 loss results match ours, but it would be interesting to try to generalize our results to other losses, and we may explore this in future work.  Also, the effects of depth on training speed are deeply intertwined with initialization strength.  For initialization strengths that optimize generalization (i..e. small weights), depth would slow down training for L2 loss at least. However, for large initial weights (which won’t help with regularization), depth can speed up convergence of the training error.  All this can be shown analytically with deep linear networks, but we feel these issues are slightly peripheral to the scope of this current paper.
>
> (continued below)

---

> > ### Author Response · Authors · 2018-10-18
> > **responses and answers (continued)**
> >
> >
> >
> > - "It seems to me that in your analysis having a more over-parametrized student (larger N2) can never be beneficial for final test error. This seems to contradict the results in this paper on the effects of over-parametrization on test error (over-parametrization improves test error): https://arxiv.org/abs/1805.12076"
> >
> > Yes, in the linear setting over-parameterization cannot decrease test error when N2 (number of student hidden units) is larger than N2bar (number of teacher hidden units). However, if N2 is less than N2bar, increasing N2 will decrease optimal early stopping test error (as well as training error at late times).  The main point of our paper is that in the linear setting, overparameterization does not hurt, or increase, the test error even without regularization, at the optimal early stopping time.
> >
> > We speculate that the intuitive reason why in nonlinear networks test error continues to reduce as the network size increases beyond the size at which zero training error is possible, is that the actual ground truth function to be learned may lie outside the expressive capacity of even large networks, and so larger expressive capacity may help with generalization, as long as optimization has some implicit inductive bias towards the ground truth function. In essence, perhaps the larger nonlinear network can cobble together features needed to match the ground truth function if it has more features, or neurons to play with.  In the linear setting this does not occur because as soon as the rank of the student exceeds that of the teacher, the teacher’s ground truth function lies within the expressive capacity of the student.
> >
> > Again, the main point of our paper is overparameterization does not hurt in the linear setting because the gradient descent from small initial weights has an inductive bias towards low rank input-output maps at the early stopping time.  This in and of itself may be useful in shedding light on inductive biases in nonlinear networks.  However, further work in nonlinear nets, or linear student nets with a nonlinear teacher, where the ground truth function lies outside the set of functions reachable by even large student networks, would likely be needed to address the issues raised in this comment.  However, we feel that may lie outside the scope of our current study.
> >
> >  - "In practice, early stopping does not really seem to be necessary to prevent overfitting in most cases (for example, see the error curves in this paper: https://arxiv.org/abs/1512.03385). Again, this seems to contradict a main result of this paper. Could you comment on why early stopping does not seem to be needed in practice (regularization?, learning rate adaptation?)?"
> >
> > It's worth noting that we plotted log losses in order to emphasize the effects of overfitting at a range of very different SNRs. However, in the high SNR cases, a change of log test error from -4 to -3 due to overfitting will not show a dramatic effect in the (non-log) test error as plotted in the paper you reference.
> >
> > In general, the balance between generalizing versus overfitting will depend on many features we highlighted, including the complexity of the network relative to the task, the structure of the task and how noisy it is, etc., as well as features like regularization that (as you mention) we have not explored yet. The results you note likely depend on many of these features, and we think our theory will help to parse out why overfitting is more problematic in some tasks than others.  For example, regularization could alleviate the need to precisely find an optimal early stopping time, and as we raised above, it is possible to extend this work to L2 regularization.
> >
> > - "For the non-linear results, I'm curious why you chose the leaky-relu? [...]"
> >
> > Results with relu look similar. We chose the leaky relu because it can output negative values, which made it easier to compare on the linear tasks we used for the linear networks (relu could not achieve 0 training error). Leakage is only 0.1, so it is relatively close to ReLu.
> >
> > - "[…] how do you envision explaining the effects of architecture with the kind of analysis presented in this paper? ”
> >
> > Many of these choices could potentially be seen as biases toward certain statistical structure in data that you would then preferentially learn. However, we do not have quantitative theory for all the different nonlinear architectural choices made in deep learning. Some are better understood than others however.   For example, convolution is relatively well understood: if the statistical structure of your data obeys a symmetry (i.e. translation invariance for images) then it is worthwhile to build this symmetry into the network to reduce sample complexity.  It may be possible to generalize our work to linear convolutional networks for example.

---

### Meta-Review · Area_Chair1 · 2018-12-16
**interesting result**

**Confidence:** 4
**Recommendation:** Accept (Poster)

**Metareview:**

The authors provide a new analysis of generalization in deep linear networks, provide new insight through the role of "task structure". Empirical findings are used to cast light on the general case. This work seems interesting and worthy of publication.